# Chemically Defined, Efficient Megakaryocyte Production from Human Pluripotent Stem Cells

**DOI:** 10.3390/cells14221835

**Published:** 2025-11-20

**Authors:** Jae Eun Kim, Yeonmi Lee, Yonghee Kim, Sae-Byeok Hwang, Yoo Bin Choi, Jongsuk Han, Juyeol Jung, Jae-woo Song, Je-Gun Joung, Jeong-Jae Ko, Eunju Kang

**Affiliations:** 1CHA R&D Institute, CHA Bundang Medical Center, Seongnam 13488, Republic of Korea; je6740@chamc.co.kr (J.E.K.); yeonmilee82@chamc.co.kr (Y.L.); dydgml@chamc.co.kr (Y.K.); sbhwang@chamc.co.kr (S.-B.H.); choiyb88@chamc.co.kr (Y.B.C.); highko@cha.ac.kr (J.-J.K.); 2Department of Biomedical Science, College of Life Science, CHA University, Seongnam 13488, Republic of Korea; flvm1023@gmail.com (J.H.); jgjoung@cha.ac.kr (J.-G.J.); 3CHA R&D Institute, Artificial Intelligence Omics Research Center, Seongnam 13488, Republic of Korea; juyeol@chamc.co.kr; 4Department of Laboratory Medicine, Yonsei University, College of Medicine, Seodaemoon-gu, Seoul 03722, Republic of Korea; labdx@yuhs.ac.ac

**Keywords:** pluripotent stem cells, megakaryocyte differentiation, Butyzamide, M-CSF, 3D suspension culture

## Abstract

**Highlights:**

**What are the main findings?**
Butyzamide (MPL agonist) plus M-CSF efficiently generates hPSC-derived MKs, replacing TPO.

**What is the implication of the main finding?**
PSC-MKs can be used as a source for disease modeling, mechanistic studies, and in vitro platelet production.

**Abstract:**

Platelet shortage poses a significant barrier to research and transfusion therapies because native megakaryocytes (MKs) are scarce in blood. To overcome this limitation, pluripotent stem cell–derived MKs (PSC-MKs) offer a standardized, donor-independent platform for research and therapeutic development, including disease modeling and ex vivo platelet production. Here, we report a chemically defined, feeder-free protocol to generate MKs from human pluripotent stem cells (hPSCs). The protocol combines the small molecule MPL agonist Butyzamide, macrophage colony-stimulating factor (M-CSF), and three-dimensional (3D) suspension culture, achieving high efficiency and reproducibility. Butyzamide replaced recombinant thrombopoietin (TPO), yielding comparable CD41^+^/CD42b^+^ populations and enhanced polyploidization. M-CSF accelerated nuclear lobulation and induced 4N MKs, while 3D culture increased yield, cell size, and substrate detachment. Multiple independent assays confirmed mature MK hallmarks, multi-nuclei, demarcation membranes, granules, and elevated mitochondrial respiration. Single-cell RNA sequencing outlined a continuous trajectory from early progenitors to functionally specialized MK subsets. This platform enables reliable MK supply for mechanistic studies and in vitro platelet production, advancing both basic research and therapeutic development.

## 1. Introduction

Megakaryocytes (MKs) are the bone marrow-resident progenitors responsible for platelet biogenesis, yet they comprise less than 0.1% of peripheral blood nucleated cells, severely limiting their direct isolation for mechanistic studies, drug screening, and therapeutic applications [1,2]. Given the critical importance of platelets in hemostasis, vascular integrity, and tissue repair, and their clinical indispensability for treating thrombocytopenia, supporting chemotherapy, and managing hemorrhagic conditions [3,4]. There is an urgent need for alternative MK sources that circumvent donor dependence, short shelf life, and alloimmunization risks associated with transfusion products [5,6]. Human pluripotent stem cells (hPSCs), including embryonic stem cells (ESCs) and induced pluripotent stem cells (iPSCs), offer an inexhaustible reservoir for MK derivation and in vitro platelet manufacturing [7,8].

However, prevailing differentiation protocols rely heavily on recombinant thrombopoietin (TPO) to activate the MPL receptor and drive MK proliferation, endomitosis, and cytoplasmic maturation [9,10]. These TPO-based methods not only incur high production costs and exhibit lot-to-lot variability, but also often achieve only modest differentiation efficiencies, limiting overall MK yield and throughput. Such limitations pose substantial barriers to scalable, Good Manufacturing Practice (GMP)-compliant workflows [7,11,12].

To address these limitations, we first evaluated Butyzamide, a non-peptidyl, small-molecule MPL agonist that can be synthesized at scale and remains chemically defined and batch-consistent. Butyzamide provides potent and sustained MPL activation at a fraction of the cost of recombinant TPO, driving MK emergence and robust polyploidization from hPSC-derived progenitors without compromising viability or differentiation fidelity [13,14]. Next, we leveraged three-dimensional (3D) suspension culture to recapitulate the complex bone marrow niche, enabling physiologically relevant cell–cell and cell–matrix interactions, dynamic nutrient and cytokine gradients, and enhanced cytoplasmic expansion and proplatelet formation [15]. In contrast to two-dimensional monolayers, our optimized 3D system markedly increases MK yield, cell size, and the proportion of highly polyploid cells, reflecting accelerated maturation kinetics. Finally, we incorporated macrophage colony-stimulating factor (M-CSF) to reconstitute critical hematopoietic niche support. M-CSF engages CSF1R on macrophage-like support cells, which secrete paracrine factors (e.g., CXCL12, IL-6) and mediate juxtacrine VCAM-1/VLA-4 interactions that stabilize MK progenitors and promote endomitosis [16,17]. M-CSF supplementation accelerates differentiation kinetics, enhances high-ploidy MK emergence, and reduces batch-to-batch variability. We hypothesized that the sequential integration of Butyzamide-mediated MPL activation, 3D suspension culture, and M-CSF-driven niche support would synergistically drive efficient, cost-effective, and scalable MK production from hPSCs. Here, we describe the design and validation of this chemically defined platform, demonstrating its utility for both fundamental thrombopoiesis research and the ex vivo manufacture of platelets for clinical translation.

## 2. Materials and Methods

### 2.1. Cell Culture and Differentiation

Human pluripotent stem cells (hPSCs) were provided by CHA University (Seongnam, Republic of Korea) under institutional review board approval (IRB no. 1044308-202105-LR-025-08). All differentiation experiments were conducted under research laboratory conditions (Table 1). Procedures were documented carefully to ensure reproducibility and traceability.

hPSCs were cultured on vitronectin-coated 6-well plates in StemMACS medium, supplemented with 10 µM Fasudil for the first 2 days. Fasudil, a ROCK inhibitor, was used to improve cell survival during the early differentiation phase [18]. For hematopoietic progenitor cell (HPC) differentiation, cells were treated with 10 µM CHIR-99021 in StemPro34 medium for 2 days. From days 2 to 4, the medium was supplemented with 50 ng/mL human bone morphogenetic protein 4(BMP-4), 75 ng/mL human vascular endothelial growth factor (VFGF), and 50 ng/mL fibroblast growth factor 2(bFGF). From days 4 to 5, the medium was further supplemented with 50 ng/mL bFGF, 75 ng/mL VEGF, 10 µM SB431542, and 1 µM retinoic acid. From days 5 to 9, the medium contained 50 ng/mL stem cell factor (SCF) and 10 ng/mL bFGF. From days 9 to 11, the medium included 50 ng/mL SCF, 10 ng/mL interleukin 3 (IL-3), and 1 mM valproic acid (VPA). From days 5 to 11, 0.1% polyvinyl alcohol (PVA) was added to the medium. After 16 days of HPC differentiation, HPCs were transferred to IMDM medium supplemented with 5 U/mL heparin (Sigma-Aldrich, St. Louis, MO, USA, #H3149-100KU) to initiate MK differentiation. Butyzamide was introduced from the second week of HPC culture and maintained during the entire MK differentiation process.

For the 2D differentiation protocol, HPCs were continuously cultured on vitronectin-coated 6-well plates throughout the entire differentiation process under static conditions For the 3D suspension culture, HPCs at day 11 of differentiation were transferred into vented-cap 125 mL Erlenmeyer flasks (Corning Inc., Corning, NY, USA, CL431143) containing IMDM-based MK differentiation medium and cultured on an orbital shaker (N-BIOTEK, Bucheon, Gyeonggi-do, Republic of Korea, NB-101SRC) at 75 rpm to promote cell aggregation and suspension growth. In the first week of MK differentiation, the medium was supplemented with 20 ng/mL bFGF, 10 ng/mL SCF, 10 ng/mL IL-3, and 10 µM Fasudil with cells cultured under shaking conditions (75 rpm). Cells were maintained under dynamic suspension conditions to promote aggregation and mimic aspects of the bone marrow microenvironment [15]. In the second week, the medium was supplemented with 20 ng/mL bFGF, 10 nM Butyzamide, 10 µM Fasudil, and 70 nM. From week 3, the medium was supplemented with 20 ng/mL bFGF, 10 nM Butyzamide, 10 µM Fasudil, and 5 ng/mL M-CSF to continue the differentiation process. Half-medium changes were performed every 3 days.

### 2.2. Flow Cytometry Analysis

Suspended cells were harvested and washed with 2% FACS buffer (PBS + 2% FBS), followed by staining with the following antibodies for 30 min at 4 °C in the dark: anti-human CD41-APC (BioLegend, San Diego, CA, USA, #303710) and anti-human CD42b-PE (eBioscience, San Diego, CA, USA, #12-0429-42). After washing, cells were analyzed using a CytoFLEX flow cytometer (Beckman Coulter, Brea, CA, USA), and data were processed with FlowJo v10.6 software (Treestar Inc., Ashland, OR, USA).

### 2.3. Ploidy Analysis

Cells were fixed in cold 70% ethanol for 15 min at 4 °C, washed, and treated with 50 µL RNase A solution (100 µg/mL, Infusion Tech, Inc., Miramar, FL, USA, #IFT-RA01), followed by 500 µL propidium iodide (PI) solution (50 µg/mL, Invitrogen, Waltham, MA, USA, #V13241). After incubation for at least 30 min at 4 °C on a shaker, samples were analyzed by flow cytometry.

### 2.4. Immunofluorescence Staining

PSC-derived MKs (week 4) were fixed with 4% paraformaldehyde (Biosesang, Seongnam, Republic of Korea, #P2031), permeabilized using 0.1% Triton X-100 (Sigma-Aldrich, St. Louis, MO, USA, #T8787), and incubated overnight at 4 °C with primary antibodies against β-tubulin (TUBB1, Abcam, Cambridge, UK, #ab179513) and α-tubulin (TUBA1A, Abcam, Cambridge, UK, #ab7291). After washing, cells were incubated with Alexa Fluor-conjugated secondary antibodies (Goat Anti-Rabbit IgG H&L Alexa Fluor 555, Abcam, Cambridge, UK, #ab150078; 1:500; Goat Anti-Mouse IgG H&L Alexa Fluor 488, Abcam, Cambridge, UK, #ab150113; 1:1000) for 1 h at room temperature. Nuclei were counterstained with DAPI (Invitrogen, Waltham, MA, USA, #D1306), and images were captured using a fluorescence microscope (Olympus, Tokyo, Japan, Model IX73).

### 2.5. Transmission Electron Microscopy (TEM)

PSC-derived MKs at week 3 were fixed with 2.5% glutaraldehyde, post-fixed in 1% osmium tetroxide, embedded in epoxy resin, sectioned at 70 nm, and stained with uranyl acetate and lead citrate. Imaging was performed using a JEOL JEM-1400 TEM (JEOL Ltd., Akishima, Tokyo, Japan).

### 2.6. Mitochondrial Respiration Analysis

Mitochondrial respiration assays were performed following a previously described protocol [18]. For the measurement of mitochondrial respiration, PSC-derived MKs at weeks 2, 3, and 4 were suspended in running medium, seeded onto Poly-D-lysine–coated Seahorse XFe96/XF Pro cell culture microplates (Poly-D-lysine, Sigma-Aldrich, St. Louis, MO, USA, #A-003-E; Agilent, Santa Clara, CA, USA, #103794-100), centrifuged (200× *g*, 1 min, zero break) to promote cell attachment, and incubated for 30 min in a 37 °C non-CO_2_ incubator. The running medium was prepared by dissolving DMEM powder (Sigma-Aldrich, St. Louis, MO, USA, #D5030) in sterile triple-distilled water and supplementing it with D-glucose (Sigma-Aldrich, St. Louis, MO, USA, #G6152), sodium pyruvate (Sigma-Aldrich, St. Louis, MO, USA, #P5280), and L-glutamine (Gibco, Grand Island, NY, USA, #21051024). Mitochondrial respiration was assessed using a Seahorse XF Pro Analyzer (Agilent, Santa Clara, CA, USA). The drug compounds (Agilent, Santa Clara, CA, USA, #103015-100) were used at final concentrations of 1.5 µM oligomycin, 2 µM FCCP, and 0.5 µM rotenone/antimycin A. After analysis, the cells were harvested, and DNA was extracted using 1× lysis buffer. The DNA concentration was measured and used for normalization.

### 2.7. Single-Cell RNA Sequencing

Single-cell RNA sequencing was conducted by the Advanced Omics Research Center, CHA Future Medicine Research Institute (Seongnam, Republic of Korea). MK samples were prepared in-house and transferred to the center for library preparation, sequencing, and data analysis. Processed datasets and cluster annotations were provided for interpretation. Accession numbers will be provided upon manuscript acceptance or prior to publication.

### 2.8. Quantification and Statistical Analysis

Quantified data were analyzed using GraphPad Prism software (version 10.0; GraphPad Software, San Diego, CA, USA). Statistical comparisons between two groups were performed using unpaired Student’s *t*-test. For comparisons among more than two groups, one-way ANOVA followed by appropriate post hoc tests (e.g., Tukey’s multiple comparisons test) was used. A *p*-value of < 0.05 was considered statistically significant. Significance levels were indicated as follows: ns, *p* > 0.05; *, 0.01 < *p* ≤ 0.05; **, 0.001 < *p* ≤ 0.01; ***, 0.0001 < *p* ≤ 0.001; ****, *p* ≤ 0.0001

All differentiation experiments were performed in three independent experimental series (n = 3), each starting from separate PSC cultures. Each series was subjected to the same differentiation protocol and analyzed independently. Data are presented as mean ± SD unless otherwise indicated.

## 3. Results

### 3.1. Butyzamide Enhances PSC-Derived Megakaryocyte Differentiation

To evaluate strategies for improving megakaryocyte (MK) differentiation efficiency from pluripotent stem cell (PSC)-derived hematopoietic progenitor cells (HPCs), we compared our standard thrombopoietin (TPO)-based differentiation protocol with an alternative approach using Butyzamide, a non-peptidyl, small-molecule agonist of the MPL receptor (Figure 1A). Notably, Butyzamide is readily synthesized and approximately ten times less expensive than recombinant TPO [10,13,19]. Hematopoietic progenitor cells (HPCs) were generated from pluripotent stem cells (PSCs) between days 14 and 18, then differentiated into megakaryocytes (MKs) through week 5. Morphological evaluation at weeks 2–4 showed comparable cell size and cytoplasmic features in both TPO- and Butyzamide-treated cultures (Figure 1B). While TPO-treated cells displayed higher CD42b expression during early differentiation (weeks 3–4), Butyzamide ultimately yielded equivalent CD42b^+^ MKs by week 5.

Flow cytometry analysis of CD42b expression revealed proportions of CD42b^+^ cells at weeks 3, 4, and 5 of 78% vs. 61%, 89% vs. 76%, and 94% vs. 95% in TPO- and Butyzamide-treated groups, respectively, indicating a trend toward higher CD42b levels in the TPO group at weeks 3 and 4 that converged by week 5 (Figure 1C). By week 5, CD41^+^/CD42b^+^ double-positive MKs comprised 88% of both cultures, with both treatments achieving >95% expression of the mature marker CD42b (Figure 1D). Ploidy analysis further revealed a tendency for Butyzamide-treated cultures to exhibit a higher proportion of ≥4N cells (69% vs. 56%), suggesting enhanced endomitosis and cytoplasmic expansion in these cells (Figure 1E). Cells were stratified into small, intermediate, and large subsets based on forward scatter (FSC) and side scatter (SSC) gating, and subsequent ploidy analysis demonstrated a clear trend toward higher DNA content in the larger cell populations. This result further supports the superior maturation effect of Butyzamide on MK differentiation.

### 3.2. Three-Dimensional Suspension Culture Enhances PSC-Derived Megakaryocyte Differentiation

Building on our two-dimensional (2D) monolayer findings and guided by reports of improved megakaryocyte yields in 3D culture platforms [19,20,21]. We applied a 3D suspension system to direct differentiation of PSCs into megakaryocytes (Figure 2A). Suspension culture conditions were optimized by testing different working volumes and shaking speeds. The optimal condition was identified as 20 mL at 75 rpm, which was subsequently used for all experiments. The cytokine treatment schedule was identical to the 2D protocol, confirming that differences in differentiation efficiency were primarily due to culture configuration rather than cytokine exposure.

Under these dynamic conditions, floating hematopoietic progenitor cells (HPCs) remained in suspension and freely extended in all directions, creating a three-dimensional (3D) dynamic environment distinct from the static, planar configuration of 2D monolayer culture [15]. By weeks 3–4, cells in 3D formed densely packed clusters exhibiting more pronounced cytoplasmic expansion than those in 2D (Figure 2B). Furthermore, 3D cultures exhibited minimal cell adhesion to the culture substrate, in contrast to 2D cultures, where a substantial number of cells remained adherent to the plate surface. Comparison of 2D and 3D culture systems demonstrated significantly higher MK differentiation in 3D conditions, with CD42b^+^ cells comprising 61% vs. 68% at week 3 and 75% vs. 89% at week 4 (*p* < 0.05 for both, Figure 2C). Flow cytometry demonstrated that 3D culture achieved 80.2% CD42b^+^ MKs vs. 60.7% in 2D at week 4 (Figure 2D). Additionally, cells in 3D culture tended to exhibit larger cell size compared to those in 2D culture.

DNA ploidy analysis was conducted specifically on CD42b^+^ MKs, and revealed a higher proportion of mature polyploid MKs, 68% 4N, 20% 8N, and 8% ≥16N, compared to 87% 4N, 9% 8N, and virtually no ≥16N cells in 2D (Figure 2E). As expected, ploidy increased in a size-dependent manner: the large FSC/SSC-defined subset exhibited the highest proportion of ≥4N and ≥8N cells, followed by the intermediate and small subsets.

These data confirm that 3D suspension culture synergizes with Butyzamide-activated MPL signaling to accelerate endomitosis and cytoplasmic maturation, offering a scalable, high-yield platform for economical platelet production.

### 3.3. M-CSF Supplementation Accelerates PSC-Derived Megakaryocyte Differentiation

The impact of M-CSF on PSC-derived megakaryocyte differentiation was evaluated by assessing its effects on progenitor expansion, micro-environmental conditioning, and CSF1R signaling within our 3D PSC-MK platform. M-CSF is a central hematopoietic regulator that promotes survival and differentiation of monocyte/macrophage lineages and reshapes the bone marrow niche through both paracrine signaling and direct cell–cell interactions [20,21,22]. It was hypothesized that M-CSF-driven generation of macrophage-like support cells would accelerate megakaryocyte commitment and enhance poly-ploidization by potentiating MPL-JAK2-STAT5 activation [23,24,25]. Indeed, whereas without M-CSF, culture required 4 weeks to reach approximately over 90% CD42b^+^ MKs (Figure 2D), M-CSF-supplemented cultures achieved comparable CD42b^+^ frequencies within 4 weeks (Figure 3A). Importantly, although the final differentiation efficiency was comparable between M-CSF-treated and untreated groups at weeks 4 the use of M-CSF markedly accelerated the differentiation timeline, improved the consistency of maturation kinetics, and Significantly reduced batch-to-batch variation (Figure 3B). DNA ploidy analysis further revealed a pronounced increase in highly polyploidy cells, including both ≥16N and a distinct 32N population, observed exclusively in the M-CSF-treated cultures (Figure 3C), with quantification presented (Figure 3D). To evaluate functional maturation, mitochondrial respiration assays were performed across differentiation stages at weeks 2, 3, and 4.

The maximal oxygen consumption rate (OCR) progressively increased over time, with the highest levels observed at weeks 4, indicating enhanced mitochondrial reserve capacity during MK maturation (Figure 3E). This trend is consistent with previous reports suggesting that increased mitochondrial oxidative capacity is crucial for megakaryocyte differentiation and efficient platelet production [26,27,28]. Comparative analysis of mitochondrial respiration at week 4 demonstrated a significant increase in OCR in cultures differentiated with M-CSF compared to those without M-CSF (Figure 3F), suggesting that M-CSF facilitates mitochondrial activation associated with late-stage megakaryocytic maturation. Taken together, these results indicate that M-CSF supplementation not only expedites megakaryocytic differentiation and enhances cytoplasmic maturation but also improves overall differentiation robustness and reproducibility. By recreating key macrophage-megakaryocyte interactions present in the native bone marrow niche, M-CSF enables more efficient, faster, and more reliable production of platelet-competent MKs in scalable 3D bioprocesses.

### 3.4. Multiple Assays Confirmed That the Cells Are PSC-Derived Megakaryocytes

To comprehensively validate the identity and functional maturation of PSC-derived MKs under the Butyzamide-M-CSF-3D culture regimen, PSC-derived HPCs were cultured for 2 to 4 weeks. Morphological assessment by light microscopy revealed classic megakaryocytic features, including multi-lobed nuclei, proplatelet-like extensions, and an elaborated demarcation membrane system by weeks 3 (Figure 4A). Live-cell tomographic imaging using Nano-live demonstrated abundant cytoplasmic organelles and well-developed demarcation membranes (Figure 4B). Transmission electron microscopy (TEM) analysis revealed a dense and extensive demarcation membrane system and abundant α-granules, with ultrastructural features closely resembling those of human primary MKs (Figure 4C). Immunocytochemical analysis of differentiated cultures demonstrated robust β1-tubulin (TUBB1) staining along cytoskeletal filaments, granular PF4 localization within the cytoplasm, and punctate 5-HT signals, all coexisting within cells exhibiting multi-lobed, multinucleated nuclei (DAPI) characteristic of mature MKs (Figure 4D). Flow cytometric analysis further characterized the cellular composition of the differentiated cultures, revealing that 94% of cells expressed β1-tubulin (TUBB1) and 69% expressed PF4 compared with unstained controls (Figure 4E). These results demonstrate that the majority of cells acquired megakaryocytic identity while maintaining a degree of heterogeneity within the differentiated population. Western blot analysis was performed using lysates from PSC-derived MKs and primary human MKs (Figure 4E). PSC-derived MKs expressed megakaryocyte-specific proteins β1-tubulin and PF4, along with the housekeeping protein β-actin. These results confirm the presence of key structural and granule-associated MK proteins indicative of mature MK identity.

Together, these data confirm cytoskeletal reorganization, granule biogenesis, and nuclear poly-ploidization consistent with advanced megakaryocyte maturation. Collectively, these multiple independent analyses validate that PSC-derived MKs differentiated under the Butyzamide-M-CSF-3D culture system achieve ultrastructural, molecular, and functional maturation comparable to primary human MKs, supporting the feasibility of this approach for efficient and large-scale platelet production.

### 3.5. Single-Cell Transcriptomic Validation of Efficient Megakaryocytic Differentiation

To further characterize the fidelity and progression of megakaryocytic differentiation at single-cell resolution, we performed single-cell RNA sequencing (scRNA-seq) on PSC-derived MKs differentiated using the Butyzamide–M-CSF–3D protocol for three weeks. Morphological assessment confirmed the presence of large, multinucleated, and rounded cells typical of maturing MKs (Figure 5A). Flow cytometry analysis supported these observations, revealing efficient differentiation, with 98% of cells expressing both CD41 and CD42b, while CD11b and CD14 showed minimal expression (Figure 5A). Following quality control, 11,978 cells were retained for downstream scRNA-seq analysis. Unsupervised clustering identified ten transcriptionally distinct populations (Clusters 0–9), with Clusters 0 (24.1%) and 1 (22.7%) being most abundant, followed by Clusters 2 (19.4%), 3 (9.5%), and others ranging from 7.6% (Cluster 4) to 0.7% (Cluster 9) (Figure 5B). Dimensionality reduction using Uniform Manifold Approximation and Projection (UMAP) and t-distributed Stochastic Neighbor Embedding (tSNE) revealed ten transcriptionally distinct clusters corresponding to specific MK subtypes (Figure 5C). UMAP revealed a clear differentiation trajectory beginning with early-transition progenitors (Cluster 6), progressing through primitive MK progenitors (Cluster 9) and early committed MK progenitors (Cluster 8), into myeloid-associated MK populations (Cluster 4). Subsequent branches diverged into specialized MK subsets, including immune-activated MKs (Cluster 0), inflammation-associated progenitors (Cluster 1), immune-featured MKs (Cluster 2), antigen-presenting progenitors (Cluster 3), regulatory MKs (Cluster 5), and immune-modulatory MKs (Cluster 7) (Figure 5C). To define transcriptional signatures across MK maturation, we analyzed the expression of established early and late MK-associated genes across clusters (Figure 5D). Early hematopoietic and MK lineage markers (ETV6, RUNX1, IGF1, FLI1) were broadly expressed, confirming lineage commitment. Critical early MK transcription factors (GATA1, GATA2) were strongly enriched in Clusters 8 and 9, indicative of lineage specification. In contrast, late-stage maturation markers (PLEK, TUBB1, SELP) exhibited peak expression in terminally differentiated clusters, consistent with advanced cytoskeletal remodeling and granule biogenesis. To further refine MK subtype classification, representative differentially expressed genes (DEGs) that most contributed to cluster segregation were visualized using violin plots (Figure 5E). These DEGs highlighted distinct transcriptional features defining each cluster’s identity and maturation stage.

Collectively, these data illustrate a dynamic and branched MK differentiation landscape, spanning early progenitors to mature functional subsets. The consistent and widespread expression of canonical MK markers across distinct transcriptional clusters confirmed efficient and robust megakaryopoiesis. These single-cell transcriptomic analyses thus provide high-resolution validation of our differentiation platform’s capacity to recapitulate physiologically relevant MK maturation trajectories reproducibly.

## 4. Discussion

Large-scale production of MK from human pluripotent stem cells (PSCs) is hindered by the high cost and variability of recombinant thrombopoietin (TPO)-based protocols, modest differentiation efficiencies, and the inability of 2D cultures to recapitulate the bone marrow niche [7,11,29]. To overcome these limitations, we developed a chemically defined, feeder-free protocol integrating three key elements in sequence: (1) replacement of recombinant TPO with Butyzamide, a small-molecule MPL agonist; (2) three-dimensional (3D) suspension culture to mimic marrow architecture; and (3) supplementation with macrophage colony-stimulating factor (M-CSF) for niche support.

Butyzamide proved a cost-effective alternative to TPO, sustaining robust MPL signaling that drove efficient colony-forming unit-MK emergence, enhanced proliferation, and elevated endomitosis. By weeks 5, Butyzamide-treated cultures matched TPO controls in CD41^+^/CD42b^+^ expression and exhibited a higher proportion of ≥4N MKs, underscoring its capacity to promote functional maturation without the expense and batch variability of recombinant proteins [7,13,19]. Transitioning to 3D suspension culture markedly improved MK yield and maturation compared to 2D monolayers.

The 3D format fostered physiologically relevant cell–cell and cell–matrix interactions, resulting in larger cell size, reduced substrate adhesion, and a significant increase in highly polyploid (≥16N) MKs. These enhancements align with previous reports that 3D systems accelerate cytoplasmic expansion and proplatelet formation by recapitulating dynamic cytokine gradients and mechanical cues present in the marrow niche [11,15,20]. Finally, M-CSF supplementation accelerated differentiation kinetics and improved robustness.

M-CSF-driven macrophage-like support cells engaged CSF1R signaling to secrete paracrine factors and mediate juxtacrine interactions, which stabilized progenitors, promoted early nuclear lobulation, and yielded unique 32N MK populations. Batch-to-batch variability was also markedly reduced, demonstrating that macrophage-MK-MK crosstalk is critical for consistent, high-efficiency maturation [30,31].

All differentiated cells expressed canonical MK markers and lacked macrophage-specific genes, confirming their megakaryocytic identity. At the protein level, over 80% of cells were CD41^+^/CD42b^+^ double positive, indicating that the cultures predominantly consisted of MKs. MK subsets were defined based on differential expression of cluster-enriched genes in addition to shared MK markers (Figure 5E), reflecting functional diversity among MKs rather than contamination by macrophages.

Although distinct macrophage clusters were not detected in scRNA-seq analysis, CD68 expression was observed in Cluster 0, which also expressed MK-related genes. Immune MKs exhibited CD68 expression under basal conditions, supporting the physiological relevance of CD68-positive MK [32]. These findings indicate that our differentiation protocol generates MK subpopulations with immune-like characteristics, reflecting functional heterogeneity within the MK lineage. Notably, Clusters 1 and 7 also exhibited CD68 expression alongside canonical MK markers, consistent with immune-featured MKs. Classical macrophage markers CD11b and CD14 were not expressed, and protein analyses confirmed β1-tubulin (TUBB1) and PF4 expression, supporting mature MK identity. These results suggest that macrophage-like gene expression reflects functional heterogeneity within the MK lineage rather than contamination or immature progenitors.

This subset likely represents macrophage-like intermediates within the MK lineage that transiently respond to M-CSF. These hybrid populations may mediate indirect, supportive effects through paracrine or juxtacrine signaling, thereby contributing to enhanced MK maturation despite the absence of mature macrophages at the transcriptomic level.

Our multimodal validation, spanning morphological, ultrastructural, immunophenotypic, metabolic, and single-cell transcriptomic analyses, confirmed that MKs generated by the Butyzamide–3D-M-CSF platform possess hallmark features of mature, functional MKs. Transmission electron microscopy and immunocytochemistry demonstrated extensive demarcation membranes, α-granule formation, and cytoskeletal reorganization, while oxygen consumption assays revealed enhanced mitochondrial respiration consistent with terminal maturation [2,7,33]. Single-cell RNA sequencing delineated a continuous differentiation trajectory from early RUNX1^+^/ETV6^+^ progenitors to functionally specialized MK subsets expressing late-stage genes such as TUBB1, SELP, and PLEK, faithfully recapitulating in vivo ontogeny [34,35].

Moreover, our protocol generated MK subpopulations analogous not only to marrow-resident immune, regulatory, and proplatelet-forming MKs [35] but also to lung-resident MKs involved in pulmonary platelet biogenesis [30], demonstrating that the combined Butyzamide-3D-M-CSF environment effectively models both bone marrow and pulmonary niches.

In summary, the sequential integration of Butyzamide, 3D suspension culture, and M-CSF supplementation establishes a reproducible, scalable, and economically sustainable platform for PSC-derived MK production. By surmounting the cost, scale, and variability challenges of existing methods, this strategy provides a robust supply of MKs for mechanistic thrombopoiesis research and paves the way for scalable ex vivo platelet biomanufacturing in clinical settings.

## 5. Conclusions

We established a chemically defined, feeder-free platform for producing megakaryocytes from human pluripotent stem cells by integrating three elements: (i) Butyzamide-mediated MPL activation as a small-molecule substitute for recombinant TPO, (ii) three-dimensional suspension culture, and (iii) M-CSF–driven niche support. This combination enables high-efficiency, reproducible generation of PSC-MKs. The resulting cells display canonical maturation features including multinucleation, cytoplasmic granules, and elevated mitochondrial respiration. Single-cell transcriptomic trajectories corroborate these phenotypes, mapping a continuum from early progenitors to specialized MK subsets. Together, these features provide a reliable MK supply for mechanistic studies and in vitro platelet production and align with the field’s shift toward standardized, cytokine-sparing workflows that support both basic discovery and therapeutic development. Furthermore, while our study demonstrates the feasibility of producing PSC-derived megakaryocytes, it represents a prototype stage; translation to clinical-scale GMP-compliant manufacturing will require further optimization, including the use of GMP-grade reagents, process standardization, and scale-up validation. Nevertheless, this work provides a solid foundation for future translational studies aimed at therapeutic applications.

## Figures and Tables

**Figure 1 cells-14-01835-f001:**
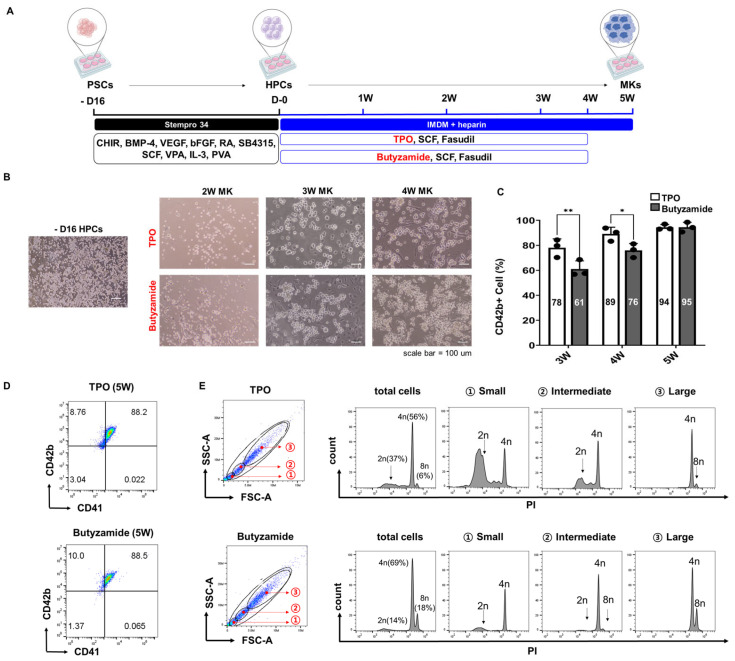
Efficient differentiation of PSCs into MKs with Butyzamide treatment. (**A**) Schematic representation of the differentiation protocol from PSCs to MKs. PSCs were differentiated into hematopoietic progenitor cells (HPCs) over 16 days and further matured into megakaryocytes (MKs) under thrombopoietin (TPO) or Butyzamide conditions. (**B**) Representative phase-contrast images show morphological changes from HPCs (D-16) to MKs at 2, 3, and 4 weeks (scale bar = 100 μm). (**C**) Bar graph showing the percentage of CD42b^+^ cells at weeks 3, 4, and 5 in TPO and Butyzamide conditions (N = 3). (**D**) Representative flow cytometry plots showing CD41 and CD42b expression in week 5 differentiated megakaryocytes under the two treatment conditions. TPO-treated cells are shown in the top row, and Butyzamide-treated cells are presented in the bottom row. (**E**) DNA ploidy analysis of week 5 megakaryocytes assessed by PI staining. Initial gating was performed based on FSC and SSC to categorize cells into small, intermediate, and large populations. Histograms display DNA content (2N, 4N, 8N, and ≥16N) within each size group. Upper plots represent the TPO condition, and lower plots correspond to Butyzamide treatment. Statistical significance was determined by two-way ANOVA. Data are presented as mean ± SD.* *p* < 0.05, ** *p* < 0.01.

**Figure 2 cells-14-01835-f002:**
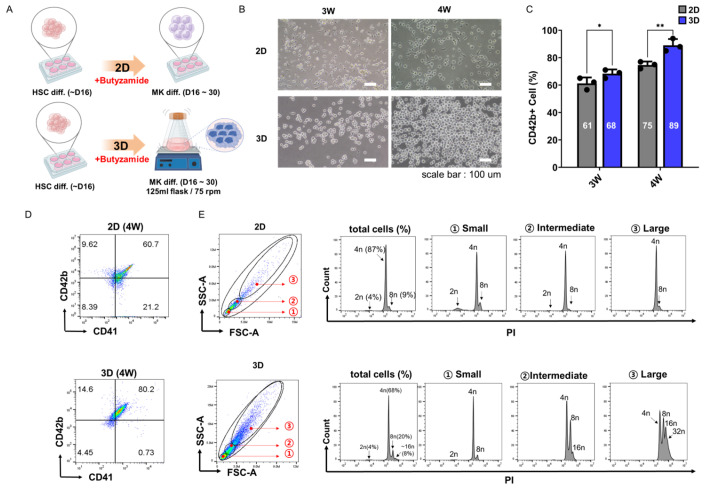
Enhanced differentiation of PSCs into MKs using a 3D culture system under Butyzamide treatment. (**A**) Schematic representation of the differentiation protocol from PSCs to MKs. PSCs were differentiated into megakaryocytes (MKs) using either 2D or 3D culture systems under Butyzamide treatment. (**B**) Representative phase-contrast images showing morphological changes in MKs at weeks 3 (2D and 3D) and weeks 4 (2D and 3D) of differentiation (scale bar = 100 μm). (**C**) Bar graph showing the percentage of CD42b^+^ cells at weeks 3 and 4 in 2D and 3D culture systems (n = 3). (**D**) Representative flow cytometry plots showing CD41 and CD42b expression in megakaryocytes differentiated for 4 weeks under Butyzamide treatment. Cells were cultured under either 2D (top row) or 3D (bottom row) conditions. (**E**) DNA ploidy analysis of 4-week differentiated CD42b^+^ megakaryocytes under Butyzamide treatment. Cells were first gated based on FSC and SSC to define small, intermediate, and large populations, followed by gating on CD42b^+^ cells for ploidy assessment. Histograms show DNA content (2N, 4N, 8N, and ≥16N) within each size group. Results from 2D cultures are shown above, and those from 3D cultures are shown below. Statistical significance was determined by two-way ANOVA. Data are presented as mean ± SD. * *p* < 0.05, ** *p* < 0.01.

**Figure 3 cells-14-01835-f003:**
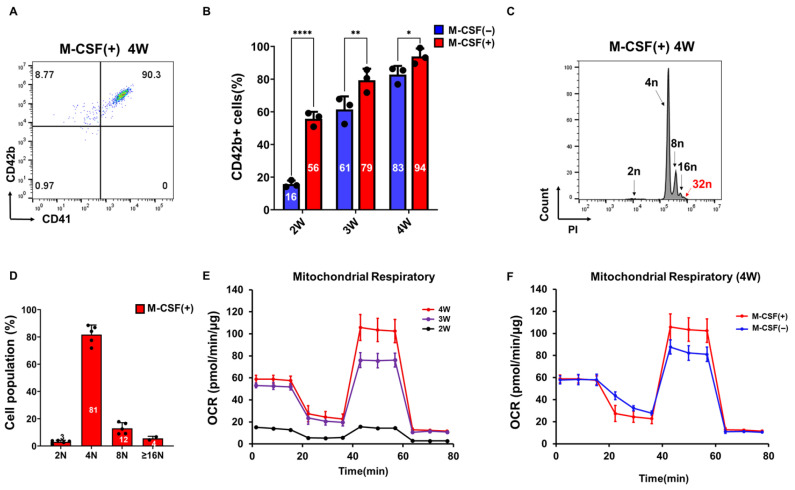
M-CSF enhances megakaryocyte differentiation and mitochondrial activity under Butyzamide treatment. (**A**) Representative flow cytometry plots showing CD41 and CD42b expression in megakaryocytes differentiated with or without M-CSF at weeks 4 under Butyzamide treatment. (**B**) Bar graphs showing the percentage of CD42b^+^ cells at weeks 2, 3, and 4 in the presence or absence of M-CSF (n = 3). (**C**) DNA ploidy analysis of megakaryocytes differentiated for 4 weeks in the presence of M-CSF, assessed by PI staining. Cells were gated into small, intermediate, and large populations based on FSC and SSC, and DNA content (2N, 4N, 8N, and ≥16N) was measured. (**D**) Quantification of DNA ploidy levels in M-CSF–treated megakaryocytes. The proportion of cells within each ploidy category is shown as mean ± SD (n = 3). (**E**) Mitochondrial respiration analysis of megakaryocytes differentiated with M-CSF at weeks 2, 3, and 4. Oxygen consumption rate (OCR) was measured using a mitochondrial stress test under M-CSF treatment. (**F**) Mitochondrial respiration analysis of megakaryocytes differentiated with or without M-CSF at week 4. Oxygen consumption rate (OCR) was measured using a mitochondrial stress test to compare mitochondrial activity between M-CSF–treated and untreated cultures. The M-CSF–treated data at week 4 in Figure 3F correspond to those presented in Figure 3E. Statistical significance was determined by two-way ANOVA. Data are presented as mean ± SD. * *p* < 0.05, ** *p* < 0.01, **** *p* < 0.0001.

**Figure 4 cells-14-01835-f004:**
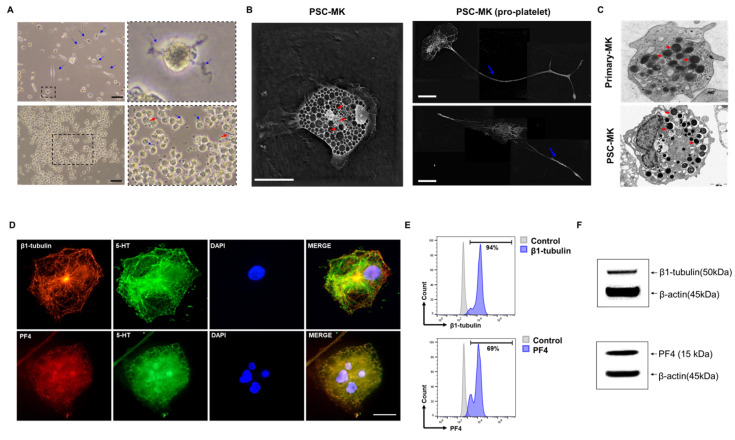
Comprehensive Multimodal Validation of PSC-Derived Megakaryocytes. (**A**) Phase-contrast microscopy image of PSC-MKs at weeks 3 showing the emergence of proplatelet extensions (blue arrows) and platelet-like particles (PLPs; red arrows) in the surrounding medium. Scale bar = 100 μm. (**B**) Label-free 3D live-cell imaging (Nanolive) visualizing intracellular α-granules (red arrows) in PSC-MKs (left), extended pro-platelet projections (blue arrows) from PSC-MKs (right). Scale bar = 20 μm. (**C**) Transmission electron microscopy (TEM) image showing ultrastructural comparison between primary MKs (top) and PSC-derived MKs (bottom). Representative TEM images comparing the ultrastructure of PSC-derived MKs and primary MKs. A dense demarcation membrane system (DMS) is present in PSC-derived MKs, indicating functional maturation. Red arrows indicate α-granules in MKs. (**D**) Immunofluorescence staining of PSC-MKs: Representative images of PSC-derived MKs after 4 weeks of differentiation. The upper images show β1-tubulin (red) and serotonin (5-HT; green), while the lower images show PF4 (red) and 5-HT (green). Nuclei were counterstained with DAPI (blue). Scale bar = 10 μm. (**E**) Flow cytometric analysis of week 3 PSC-derived MKs showing PF4 and β1-tubulin (TUBB1) expression compared with unstained controls. (**F**) Western blot analysis of week 3 PSC-derived MKs showing expression of β1-tubulin (50 kDa) and PF4 (15 kDa), with β-actin (45 kDa) as a loading control, confirming the acquisition of mature megakaryocytic features.

**Figure 5 cells-14-01835-f005:**
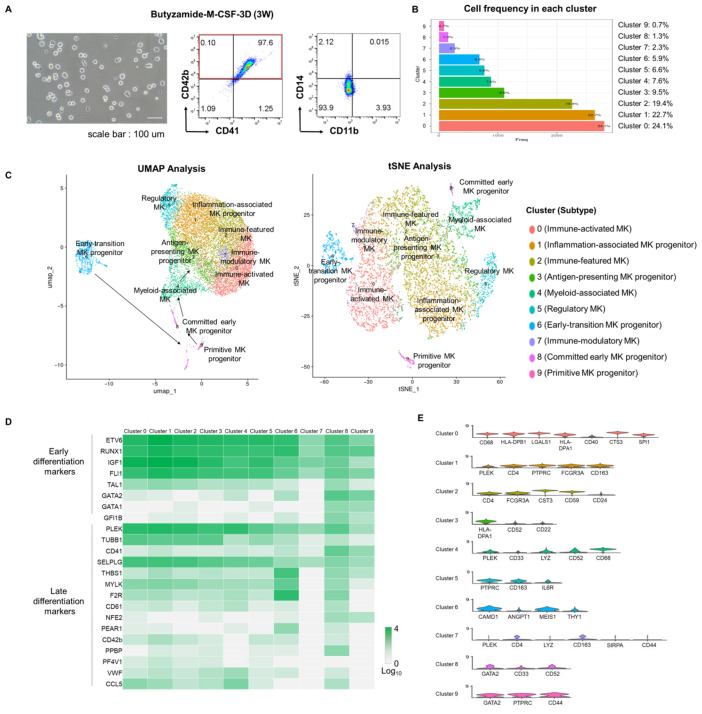
Single-Cell Transcriptomic Validation of Efficient Megakaryocytic Differentiation. (**A**) Morphology and CD marker expression of differentiated megakaryocytes used for scRNA-seq. CD41^+^CD42b^+^ cells accounted for 84.2%, confirming efficient MK induction. Additional staining for CD11b and CD14 confirmed that both macrophage-associated markers showed minimal expression, supporting the purity of the differentiated MK population. (**B**) Cell frequency across 10 clusters identified by scRNA-seq. Clusters 0, 1, and 2 were the most abundant. (**C**) UMAP and tSNE visualization of MK subtypes. The trajectory initiates from Cluster 6 (early-transition MK progenitor), progresses through Clusters 9 and 8, and branches into mature MK subsets. (**D**) Heatmap of early and late MK marker expression. MK-related genes were broadly expressed in nearly all clusters, supporting shared lineage identity. (**E**) Violin plots of representative DEGs that defined each cluster and contributed to MK subtype classification.

**Table 1 cells-14-01835-t001:** Reagents used for PSC-derived megakaryocyte differentiation.

Reagent/Material	Manufacturer	Research Grade Catalog No.	Alternative GMP Grade Catalog No.
stemMACS iPS-brew XF	Miltenyi (Bergisch Gladbach, Germany)	130-104-368	170-076-317
Stempro 34	Gibco	10640-019	A6636901
IMDM (Iscove’s Modified Dulbecco Medium)	Gibco	12440-053	12440-053
Vitronectin	Gibco	A31804	A14700
Sodium citrate	Sigma-Aldrich	S4641	S4641-25g
Fasudil HCl	AdooQ (Irvine, CA, USA)	A10381	Y-27632, TB1254-GMP (TOCRIS)
CHIR-99021	MedChemExpress (South Brunswick Township, NJ, USA)	HY-10182	TB4423-GMP (R&D systems)
BMP-4	Miltenyi	130-111-165	314E-GMP (R&D systems)
bFGF	Peprotech (Cranbury, NJ, USA)	100-18b	GMP100-18B-025
VEGF 165	Miltenyi	130-109-385	BT-VEGF-GMP (R&D systems)
Retinoic acid	Sigma (St. Louis, MO, USA)	R2625	HY-14649G (MedChemExpress)
SCF	Miltenyi	130-096-695	170-076-149
IL-3	Miltenyi	130-095-069	170-076-110
VPA	Sigma	PHR1061	1708707
Penicillin-Streptomycin, 100X	GeneDireX (Taoyuan City, Taiwan)	CC502-0100	Gentamycin, 15750060 (Gibco)
Butyzamide	Selleckchem (TX, USA)	S6988	HY-148748G (MedChemExpress)
M-CSF	Miltenyi	130-096-492	170-076-172
Human albumin 20%	SK Plasma (Andong, Gyeongsangbuk-do, Republic of Korea)	50800231	GMP Human AB Serum, HABS001-GMP-100mL (R&D systems)
PVA	Sigma-Aldrich	363146	1548065

Research-grade reagents used in this study are listed together with GMP-grade alternatives, where available.

## Data Availability

The data that support the findings of this study are available from the corresponding author.

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
