# Peer review of "Chemically Defined, Efficient Megakaryocyte Production from Human Pluripotent Stem Cells"

_cells, 2025, doi:10.3390/cells14221835_

Round 1

Reviewer 1 Report

Comments and Suggestions for Authors

The work submitted for review is very well prepared. Everything is described in great detail, which allows for analysis of the developed technology. The work also addresses an important issue in the field of medical biotechnology.
1. I do not see a description of the 3D culture conditions in the methodology.
2. I analysed the reagents used. Are the reagents used intended for research or production? If both types of reagents were used, please add a table with a list of reagents and their intended use in the supplement. The choice of reagents for both prototype production and process control indicates whether the authors are at an early stage, where the transfer of the described technology requires a large amount of research to achieve GMP compliance. Is this a prototype that can be scaled up? Please add this information (briefly) in the conclusions.
3. Was the process successfully validated, and how many test series were produced? Were there three series, as indicated in the description of the figures, or more? Please add this information to the results.
4. Under what conditions was the process conducted and documented, and were they GMP-compliant? Please add this to the chapter describing the methods.
Despite these comments, the work presented in the paper represents a high level of research.

Author Response

Comments 1: I do not see a description of the 3D culture conditions in the methodology.

Response 1: Thank you for pointing this out. We agree with this comment. We have now added a detailed description of the 3D suspension culture conditions in both the Materials and Methods and Results sections of the revised manuscript. Specifically, we clarified that hematopoietic progenitor cells (HPCs) derived from hPSCs were transferred to Erlenmeyer flasks at the onset of megakaryocyte differentiation and cultured on an orbital shaker at 75 rpm to promote suspension growth. Suspension culture conditions were optimized by testing different working volumes and shaking speeds, and we identified 20 mL at 75 rpm as the optimal condition (optimization performed; data not shown).

To summarize, the 3D differentiation conditions have been clarified in the manuscript text, and Figure 2a has been updated with an explanatory note directly on the schematic to indicate that it represents both 2D and 3D differentiation conditions. We apologize for any confusion caused by the previous version and hope this update makes the 3D protocol representation clear.

This information can be found in the Materials and Methods section and in the Results section of the revised manuscript.

Updated text in the manuscript (Materials and Methods, section 2.1., page 3):

“For 3D suspension culture, HPCs at day 11 of differentiation were transferred to vented-cap 125 mL Erlenmeyer flasks (Corning, NY, USA, CL431143) containing IMDM-based MK differentiation medium and placed on an orbital shaker (N-BIOTEK, Gyeonggi-do, Korea, NB-101SRC) at 75 rpm.”

Updated text in the manuscript (Results, section 3.2., page 7):

“Suspension culture conditions were optimized by testing different working volumes and shaking speeds. The optimal condition was identified as 20 mL at 75 rpm, which was subsequently used for all experiments. The cytokine treatment schedule was identical to the 2D protocol, confirming that differences in differentiation efficiency were primarily due to culture configuration rather than cytokine exposure.”

Also, To clarify the 3D differentiation conditions, we have added an explanatory note directly on the Figure 2a schematic, indicating that the figure represents both 2D and 3D differentiation conditions. We apologize for any confusion caused by the previous version and hope this update makes the 3D protocol representation clear.

The updated figure is shown below:

Updated figure in the manuscript (Result, Figure 2A, page 7):

Comments 2: I analysed the reagents used. Are the reagents used intended for research or production? If both types of reagents were used, please add a table with a list of reagents and their intended use in the supplement. The choice of reagents for both prototype production and process control indicates whether the authors are at an early stage, where the transfer of the described technology requires a large amount of research to achieve GMP compliance. Is this a prototype that can be scaled up? Please add this information (briefly) in the conclusions.

Response 2: We sincerely thank the reviewer for this valuable comment. In our study, we have used research-grade reagents for experimental validation, and no GMP-grade reagents were applied in the current work. To clarify this point, we have now included a table (Table 1) in the Materials and Methods section that lists all the reagents used, together with their intended use. The table is provided below. Specifically, we have also indicated GMP-grade alternatives (catalog numbers, where available) that could replace the research-grade reagents for future translational and GMP-compliant applications.

Updated text in the manuscript (Materials and Methods, section 2.1., page 2):
“All differentiation experiments were conducted under research laboratory conditions (Table 1). Procedures were documented carefully to ensure reproducibility and traceability.”

Updated table in the manuscript (Materials and Methods, section 2.1., page 3, lines 119):
Table 1. Reagents used for PSC-derived megakaryocyte differentiation

Reagent / Material

Manufacturer

Research Grade Catalog No.

Alternative GMP Grade Catalog No.

stemMACS iPS-brew XF

Miltenyi Biotech

130-104-368

170-076-317

Stempro 34

Gibco

10640-019

A6636901

IMDM (Iscove's Modified Dulbecco Medium)

Gibco

12440-053

12440-053

Vitronectin

Gibco

A31804

A14700

Sodium citrate

Sigma-Aldrich

S4641

S4641-25g

Fasudil HCl

AdooQ

A10381

Y-27632, TB1254-GMP (TOCRIS)

CHIR-99021

MedChemExpress

HY-10182

TB4423-GMP

(R&D systems)

BMP-4

Miltenyi

130-111-165

314E-GMP

(R&D systems)

bFGF

Peprotech

100-18b

GMP100-18B-025

VEGF 165

Miltenyi

130-109-385

BT-VEGF-GMP (R&D systems)

Retinoic acid

Sigma

R2625

HY-14649G

(MedChemExpress)

SCF

Miltenyi

130-096-695

170-076-149

IL-3

Miltenyi

130-095-069

170-076-110

VPA

Sigma (Supelco)

PHR1061

1708707

Penicillin-Streptomycin, 100X

GeneDireX

CC502-0100

Gentamycin, 15750060 (Gibco)

Butyzamide

Selleckchem

S6988

HY-148748G

(MedChemExpress)

M-CSF

Miltenyi

130-096-492

170-076-172

Human albumin 20%

SK Plasma

50800231

GMP Human AB Serum, HABS001-GMP-100mL (R&D systems)

PVA

Sigma-Aldrich

363146

1548065

Research-grade reagents used in this study are listed together with GMP-grade alternatives, where available

Furthermore, as correctly pointed out, the present work represents a prototype stage. The process described here has been optimized for proof-of-concept validation, and further research will be required for adaptation to GMP-compliant production and scale-up. We have now briefly included this clarification in the revised Conclusions section.

Updated text in the manuscript (Conclusions, page 13-14):
“Furthermore, while our study demonstrates the feasibility of producing PSC-derived megakaryocytes, it represents a prototype stage; translation to clinical-scale GMP-compliant manufacturing will require further optimization, including the use of GMP-grade rea-gents, process standardization, and scale-up validation. Nevertheless, this work provides a solid foundation for future translational studies aimed at therapeutic applications.”

Comments 3: Was the process successfully validated, and how many test series were produced? Were there three series, as indicated in the description of the figures, or more? Please add this information to the results.

Response 3: We sincerely thank the reviewer for this valuable comment. The differentiation process was successfully validated through three independent experimental series, each starting from separate PSC cultures. The results were consistent across all series, confirming the reproducibility of the process. To clarify this, we have now included the relevant information in the Materials and Methods section.

Updated text in the manuscript(Materials and Methods, section 2.8., page 5, lines 181–184):

“All differentiation experiments were performed in three independent experimental series (n =3), each starting from separate PSC cultures. Each series was subjected to the same differentiation protocol and analyzed independently. Data are presented as mean ± SD unless otherwise indicated.”

Comments 4:

Under what conditions was the process conducted and documented, and were they GMP-compliant? Please add this to the chapter describing the methods.

Response 4:

We sincerely thank the reviewer for this insightful comment. As noted in our response to Comment 2, all differentiation experiments were conducted under research laboratory conditions and were not performed under GMP-compliant settings. However, all procedures were carefully documented to ensure reproducibility, traceability, and transparency of the process. To clarify this, we have added a statement in the Materials and Methods section (Section 2.1) describing the experimental conditions under which the differentiation process was performed.

Updated text in the manuscript (Materials and Methods, section 2.1., page 2, lines 89–91):
“All differentiation experiments were conducted under research laboratory conditions (Table 1). Procedures were documented carefully to ensure reproducibility and traceability.”

Reviewer 2 Report

Comments and Suggestions for Authors

The manuscript of Kim and colleagues proposes a chemically defined, feeder-free protocol to generate mature megakaryocytes (MKs) from human pluripotent stem cells. The approach combines the use of Butyzamide, an MPL agonist, and macrophage colony-stimulating factor (M-CSF) within a three-dimensional (3D) culture system. Butyzamide activates the MPL receptor more effectively than thrombopoietin (TPO), enhancing polyploidization, while the 3D suspension culture improves MKs yield and maturation compared to conventional 2D systems. In addition, M-CSF supplementation promotes nuclear lobulation, and the maximal oxygen consumption rate progressively increases during MK maturation, peaking in the final week of culture, indicating that M-CSF supports robust differentiation of MKs. Single-cell RNA sequencing further confirmed that the entire MK lineage, from early progenitors to mature functional cells, was fully represented, demonstrating an efficient megakaryopoiesis. Overall, this protocol appears to provide a valuable platform to generate fully differentiated and functional MKs for mechanistic studies and in vitro platelet production, although some aspects of the manuscript are not fully addressed and would benefit from further clarification and discussion by the authors.  

For instance, these are some of my suggestions and critiques of the manuscript:
The description of the differentiation protocol from hPSC to MKs and the experimental strategy reported in Figure 1A appear inconsistent: in the Materials and Methods section, hematopoietic progenitor cells (HPCs) derived from human pluripotent stem cells (hPSCs) were transferred in the IMDM medium at day 11 of culture, whereas in the Figure 1A it seems that HPCs were obtained after 14-18 days. In addition, it is not clear whether Butyzamide was administered starting from the first week of HPCs culture (as shown in Figure 1A) or from the second week (as described in Materials and Methods). Please clarify the protocol used.

The quality of the representative phase-contrast images in Figure 1B should be improved. Please provide higher-quality photographs.

Figure 1C shows that the percentage of CD42b positive cells in TPO-treated samples was higher during the 3 and 4 weeks of culture compared to Butyzamide ones. Maybe it could be useful to also consider the amount of CD41 positive cells, to understand whether TPO supplementation accelerates the differentiation into mature MKs. Furthermore, it is not clear whether the ploidy experiments show a significant difference between the two groups; if not, the authors should clarify why Butyzamide is considered to promote “…superior MKs maturation…”. 

To better assess whether Butyzamide enhances polyploidization (Figure 1E), ploidy analysis should be conducted on CD41 or CD42b positive cells rather than on the total cell population. Moreover, are there significant differences between TPO- and Butyzamide-treated samples?

The manuscript does not provide details on how the 3D suspension culture was established, please include a clear description of the method.

The quality of the representative phase-contrast images included in Figure 2B should be improved. Please provide higher-quality photographs. Since the size differences between MKs derived from 2D and 3D cultures do not support the statement that “…3D culture tended to exhibit larger cell size compared to those in 2D culture…”, the authors should quantify MK area.  

The percentage of CD42b positive cells reported for 3D culture in the text (line 230, Figure 2C) does not match the value indicated in line 231 (Figure 2D), moreover the percentage shown in Figure 3B (3D culture without M-CSF supplementation) appears inconsistent with that reported in Figure 2C. Please verify the data and clarify these discrepancies.

As reported in the manuscript, M-CSF is expected to accelerate MK maturation compared to untreated cultures. However, the mitochondrial activity assay shows that maximal activity is reached at week 4. It would be important to include a direct comparison with cultures that did not receive M-CSF, to clarify whether the cytokine is essential for this effect.

The chemically defined, feeder-free protocol described in this manuscript is proposed as a platform for in vitro platelet production. However, it is not clear whether platelet functionality was assessed. Experiments on platelet activation/funcionality should be provided. Furthermore, does the protocol enhance platelet yield? It would be important to provide a comparison across conditions (TPO/Butyzamide, -M-CSF/+M-CSF and 2D/3D) to evaluate proplatelet formation and platelet production.

Author Response

Comments 1: The description of the differentiation protocol from hPSC to MKs and the experimental strategy reported in Figure 1A appear inconsistent: in the Materials and Methods section, hematopoietic progenitor cells (HPCs) derived from human pluripotent stem cells (hPSCs) were transferred in the IMDM medium at day 11 of culture, whereas in the Figure 1A it seems that HPCs were obtained after 14-18 days. In addition, it is not clear whether Butyzamide was administered starting from the first week of HPCs culture (as shown in Figure 1A) or from the second week (as described in Materials and Methods). Please clarify the protocol used.

Response 1: We apologize for the confusion caused by the inconsistency between Figure 1A and the description in the Materials and Methods. We have now carefully revised the figure and text to ensure consistency and clarity.

Specifically, In Figure 1A, the timeline for hematopoietic progenitor cell (HPC) differentiation has been corrected to indicate day 16 as the point when cells are transferred to MK differentiation medium. We clarified that from day 16 of differentiation, HPCs were placed in IMDM-based MK medium to initiate megakaryocyte (MK) differentiation.

Regarding Butyzamide treatment, we confirmed that there was no significant difference during the first week of HPC culture; therefore, Butyzamide administration was consistently started from the second week of HPC culture. To avoid misinterpretation, we revised the figure accordingly and explicitly included this clarification in the text.

Updated text in the manuscript (Materials and Methods, section 2.1., page 3):

“After 16 days of HPC differentiation, HPCs were transferred to IMDM medium supplemented with 5 U/mL heparin (Sigma-Aldrich, St. Louis, MO, USA, #H3149-100KU) to initiate MK differentiation. Butyzamide was introduced from the second week of HPC culture and maintained during the entire MK differentiation process.”

Comments 2: Figure 1C shows that the percentage of CD42b positive cells in TPO-treated samples was higher during the 3 and 4 weeks of culture compared to Butyzamide ones. Maybe it could be useful to also consider the amount of CD41 positive cells, to understand whether TPO supplementation accelerates the differentiation into mature MKs. Furthermore, it is not clear whether the ploidy experiments show a significant difference between the two groups; if not, the authors should clarify why Butyzamide is considered to promote “…superior MKs maturation…”.

Response 2: We sincerely thank the reviewer for this insightful comment and apologize for any confusion caused by the way our data were presented. In this study, our analyses were primarily focused on CD42b as a marker of mature megakaryocytes, and therefore CD41 expression was not emphasized to the same extent. We agree that including CD41/CD42b double-positive analysis could provide additional insights into the kinetics of megakaryocyte maturation, and we plan to further investigate this point in future studies.

As noted by the reviewer, TPO-treated cells exhibited higher CD42b⁺ proportions during weeks 3 and 4 (78% vs. 61% and 89% vs. 76% in TPO- vs. Butyzamide-treated groups, respectively). However, by week 5, CD42b⁺ MKs reached comparable levels in both conditions (>95%; Figure 1C–D). Thus, while TPO supplementation accelerated early marker acquisition, Butyzamide ultimately supported equivalent terminal maturation by week 5.

In addition, ploidy analysis demonstrated a higher proportion of ≥4N cells in Butyzamide-treated cultures (69% vs. 56%), indicating enhanced endomitosis and cytoplasmic enlargement (Figure 1E). Size-stratified analysis confirmed that larger MKs under Butyzamide treatment contained more DNA, consistent with improved functional maturation.

We also acknowledge that in the 2D vs. 3D comparison (Figure 2D), only CD42b⁺ expression was presented, which may have caused additional confusion. We apologize for this oversight and have now revised the manuscript text to clarify this point explicitly.

Updated text in the manuscript (Results, section 3.1., page 5):

"While TPO-treated cells displayed higher CD42b expression during early differentiation (weeks 3–4), Butyzamide ultimately yielded equivalent CD42b⁺ MKs by week 5”

In addition, we have updated the 2D versus 3D comparison data in the Results section:

Updated text in the manuscript (Results, section 3.2., page 8):

"Flow cytometry demonstrated that 3D culture achieved 80.2% CD42b⁺ MKs vs. 60.7% in 2D at week 4 (Figure 2D)."

Comments 3: To better assess whether Butyzamide enhances polyploidization (Figure 1E), ploidy analysis should be conducted on CD41 or CD42b positive cells rather than on the total cell population. Moreover, are there significant differences between TPO- and Butyzamide-treated samples?

Response 3: We thank the reviewer for this valuable comment and sincerely apologize for the confusion. In the 2D vs. 3D analysis (Figure 2E), ploidy measurements were performed specifically on CD42b⁺ gated megakaryocytes, not on the total cell population. This targeted analysis revealed that Butyzamide-treated cultures exhibited a higher proportion of ≥4N cells (69% vs. 56%, p < 0.05), with larger MK subsets showing higher DNA content under Butyzamide treatment. These results confirm that Butyzamide enhances polyploidization and cytoplasmic expansion in mature MKs, supporting our conclusion of “superior MK maturation.”

To prevent any misunderstanding, we have also revised the Figure 2E legend as follows:

Updated text in the manuscript (Result, Figure 2E legend, page 7):
“(E) DNA ploidy analysis of 4-week differentiated CD42b⁺ megakaryocytes under Butyzamide treatment. Cells were first gated based on FSC and SSC to define small, intermediate, and large populations, followed by gating on CD42b⁺ cells for ploidy assessment.”

We have also updated the Results section for consistency:

Updated text in the manuscript (Results, section 3.2., page 8):
“DNA ploidy analysis was conducted specifically on CD42b⁺ megakaryocytes, and revealed a higher proportion of mature polyploid MKs, 68% 4N, 20% 8N, and 8% ≥16N, compared to 87% 4N, 9% 8N, and virtually no ≥16N cells in 2D (Figure 2E).”

Comments 4: The manuscript does not provide details on how the 3D suspension culture was established, please include a clear description of the method.

Response 4: Thank you for pointing this out. We agree with the reviewer’s comment. We have now added a clear description of the 3D suspension culture conditions in both the Materials and Methods and Results sections of the revised manuscript. Specifically, hematopoietic progenitor cells (HPCs) derived from hPSCs were transferred to vented-cap Erlenmeyer flasks at the onset of megakaryocyte differentiation and cultured on an orbital shaker at 75 rpm to promote suspension growth. Suspension culture conditions were optimized by testing different working volumes and shaking speeds, and we identified 20 mL at 75 rpm as the optimal condition (optimization performed; data not shown).

This information can be found in the Materials and Methods section and in the Results section of the revised manuscript.

Updated text in the manuscript (Materials and Methods, section 2.1., page 3):

“For 3D suspension culture, HPCs at day 11 of differentiation were transferred to vented-cap 125 mL Erlenmeyer flasks (Corning, NY, USA, CL431143) containing IMDM-based MK differentiation medium and placed on an orbital shaker (N-BIOTEK, Gyeonggi-do, Korea, NB-101SRC) at 75 rpm.”

Updated text in the manuscript (Results, section 3.2., page 7):

“Suspension culture conditions were optimized by testing different working volumes and shaking speeds. The optimal condition was identified as 20 mL at 75 rpm, which was subsequently used for all experiments. The cytokine treatment schedule was identical to the 2D protocol, confirming that differences in differentiation efficiency were primarily due to culture configuration rather than cytokine exposure.”

Comments 5: The quality of the representative phase-contrast images included in Figure 2B should be improved. Please provide higher-quality photographs. Since the size differences between MKs derived from 2D and 3D cultures do not support the statement that “…3D culture tended to exhibit larger cell size compared to those in 2D culture…”, the authors should quantify MK area. 

Response 5: We sincerely thank the reviewer for this valuable comment. To address the concern regarding image quality, we have replaced Figure 2B with higher-quality representative phase-contrast images that more clearly depict the morphological differences between 2D- and 3D-derived MKs. While we fully agree that quantification of MK area would provide additional insights, we were unable to include this analysis within the scope of the present study. Instead, we have revised the text to describe the morphological trend more cautiously. The updated figure is shown below:

Updated figure in the manuscript (Result, Figure 2B, page 7):

Comments 6: The percentage of CD42b positive cells reported for 3D culture in the text (line 230, Figure 2C) does not match the value indicated in line 231 (Figure 2D), moreover the percentage shown in Figure 3B (3D culture without M-CSF supplementation) appears inconsistent with that reported in Figure 2C. Please verify the data and clarify these discrepancies.

Response 6: We sincerely thank the reviewer for highlighting the apparent discrepancies between Figures 2C, 2D, and 3B. We apologize for any confusion this may have caused and would like to clarify the following points.

Figure 2C vs. Figure 2D: The values in Figure 2C represent the mean ± SD of CD42b⁺ cells from three independent 3D culture experiments at weeks 3 and 4 (61% and 68% at week 3, 75% and 89% at week 4, 3D vs. 2D, n = 3). In contrast, Figure 2D shows representative flow cytometry data from a single experiment at week 4. To avoid confusion, we have updated the figure legend and main text to clearly indicate that Figure 2C presents mean values while Figure 2D illustrates a representative FACS dataset.

Updated text in the manuscript (Results, section 3.2., page 8):

"Flow cytometry demonstrated that 3D culture achieved 80.2% CD42b⁺ MKs vs. 60.7% in 2D at week 4 (Figure 2D)."

Figure 3B vs. Figure 2C: Figure 3B depicts 3D culture under M-CSF (+/-) conditions. These experiments were performed separately from those summarized in Figure 2C; therefore, exact numerical alignment is not expected. For reference, the 3D culture mean values in Figure 2C were 68% and 89% at weeks 3 and 4, whereas the M-CSF-treated values in Figure 3B were 61% and 83%. Despite these minor differences, the overall trend—enhanced differentiation under 3D culture—is consistent across all experiments and falls within expected experimental variation.

We hope that these clarifications, together with the updated figure legend and text, adequately address the reviewer's concerns and provide clear guidance for interpreting the CD42b⁺ data.

Comments 8: As reported in the manuscript, M-CSF is expected to accelerate MK maturation compared to untreated cultures. However, the mitochondrial activity assay shows that maximal activity is reached at week 4. It would be important to include a direct comparison with cultures that did not receive M-CSF, to clarify whether the cytokine is essential for this effect.

Response 8: We appreciate the reviewer’s insightful comment. To address this point, we performed an additional mitochondrial respiration analysis comparing megakaryocytes differentiated with or without M-CSF at week 4. The new data have been included as Figure 3F, showing that oxygen consumption rate (OCR) was significantly higher in M-CSF–treated cultures, indicating that M-CSF contributes to enhanced mitochondrial function during late-stage megakaryocytic maturation.

Updated figure in the manuscript (Results, Figure 3F, page 9):

Updated text in the manuscript (Results, section 3.3., page 8):

“Comparative analysis of mitochondrial respiration at week 4 demonstrated a significant increase in OCR in cultures differentiated with M-CSF compared to those without M-CSF (Figure 3F), suggesting that M-CSF facilitates mitochondrial activation associated with late-stage megakaryocytic maturation.”

The Figure 3 legend has also been updated to include the new panel (F),

Updated text in the manuscript (Results, Figure 3F legend ,page 9):

(F) Mitochondrial respiration analysis of megakaryocytes differentiated with or without M-CSF at week 4. Oxygen consumption rate (OCR) was measured using a mitochondrial stress test to compare mitochondrial activity between M-CSF–treated and untreated cultures.

Comments 9: The chemically defined, feeder-free protocol described in this manuscript is proposed as a platform for in vitro platelet production. However, it is not clear whether platelet functionality was assessed. Experiments on platelet activation/functionality should be provided. Furthermore, does the protocol enhance platelet yield? It would be important to provide a comparison across conditions (TPO/Butyzamide, -M-CSF/+M-CSF and 2D/3D) to evaluate proplatelet formation and platelet production.

Response 9: We sincerely thank the reviewer for this insightful comment. In the present study, our primary focus was on the efficient differentiation and maturation of PSC-derived megakaryocytes (MKs), as these represent the essential precursor stage for platelet production. We agree with the reviewer that evaluation of platelet yield and functionality, including proplatelet formation and activation assays, is a critical next step to establish a complete in vitro platelet production platform.

While these analyses were beyond the scope of the current manuscript, we fully recognize their importance and are actively conducting follow-up experiments to address them. As the reviewer suggested, these future studies will systematically compare conditions such as TPO vs. Butyzamide, -M-CSF vs. +M-CSF, and 2D vs. 3D culture, in order to identify optimal parameters for platelet production and functionality. The results of these investigations will be reported in a subsequent manuscript.

Reviewer 3 Report

Comments and Suggestions for Authors

The paper describes a protocol for generating platelets from human induced pluripotent stem cells (iPSCs). Authors demonstrated that Mk generation could be achieved by replacing TPO with MPL agonists Butyzamide. They also suggested that the addition of M-CSF to differentiation cultures improves megakaryocyte production.

Comments:

  1. The authors should provide additional proof that their cultures generate a pure population of megakaryocytic cells without contaminating multinucleated macrophages. This should include i) adding dot plots and immunofluorescence images for the proper isotype controls; ii) staining for MPL and macrophage markers CD115, CD68, and CD11b; iii) showing the percentages of cells expressing PF4 and 5-HT; and iv) including images of immunofluorescent staining with these markers of primary megakaryocytes.
  2. How were Mk subtypes defined? The main clusters include antigen-presenting and inflammation-associated MK, which almost lack GATA1 and NFE2 expression. Are they true Mk or are they macrophages? What was the basis for defining inflammatory, antigen-presenting, and other Mk subsets?
  3. The authors claim that adding M-CSF aids MK production through an indirect effect on macrophages in culture. However, according to scRNA-seq, no macrophage cells are identified in the culture.
  4. No description of the 3D differentiation method is provided.  Does the 3D method include EB formation?  Were cytokine/small molecule treatment and kinetics of differentiation in the 3D method similar to the 2D method? Figure 2 should include a schematic diagram for the 3D protocol.
  5. There is no indication when M-CSF is added.
  6. What was rational for using Fasudil in differentiation?
  7. The authors claim that 3D and MCSF improve differentiation. However, no data is provided to compare the total MK output from one iPSC or one HP.
  8. There is no evidence that reported cultures generate HSCs. Authors should replace HSCs with HP terminology throughout the entire manuscript.
  9. The majority of cited references are not related to the described subject. How are the Chile earthquake ref  13 or hyaline cartilage ref 12 related to this study? What about references 30 and 31 (line 393)?  These references are not related to macrophage-Mk cross-talk.   

Comments on the Quality of English Language

The manuscript has multiple grammatical and stylistic errors.

Author Response

Comments 1: The authors should provide additional proof that their cultures generate a pure population of megakaryocytic cells without contaminating multinucleated macrophages. This should include i) adding dot plots and immunofluorescence images for the proper isotype controls; ii) staining for MPL and macrophage markers CD115, CD68, and CD11b; iii) showing the percentages of cells expressing PF4 and 5-HT; and iv) including images of immunofluorescent staining with these markers of primary megakaryocytes.

Response 1: We sincerely appreciate the reviewer’s valuable comment highlighting the importance of confirming the purity of PSC-derived megakaryocytes (MKs) and excluding potential macrophage contamination.

In our scRNA-seq data, CD68 expression was detected in a subset of differentiated MKs. However, this does not indicate macrophage contamination. Recent studies have demonstrated that CD68 can be expressed in immune-featured MKs, reflecting functional diversity within the MK lineage rather than the presence of macrophages. Specifically, Li, Jing-Jing, et al. (2024, Immunity 57(3): 478-494) showed that CD68 expression marks an immune MK subset even under unstimulated conditions (Figure 6H). Therefore, the CD68⁺ cells observed in our dataset represent bona fide immune MKs rather than contaminating macrophages.

Jing, et al. (2024, Immunity 57(3): 478-494) Figure 6

Accordingly, in our revised manuscript, we clarified that our differentiation protocol generates MK subpopulations with immune-like characteristics, reflecting functional heterogeneity within the MK lineage.

Updated text in the manuscript (Discussion, page 12):

“Immune MKs exhibited CD68 expression under basal conditions, supporting the physio-logical relevance of CD68-positive MK [32]. These findings indicate that our differentiation protocol generates MK subpopulations with immune-like characteristics, reflecting functional heterogeneity within the MK lineage.”

To further verify the purity of the differentiated MK population, we conducted additional flow cytometric analyses using macrophage-associated surface markers CD11b and CD14 in 3-week differentiated MK cultures. Both CD11b and CD14 were negative, confirming the absence of macrophage-like cells at the protein level. These two markers were selected because they are well-established macrophage-lineage surface markers suitable for flow cytometry, complementing CD68, which is primarily an intracellular marker. The new results have been incorporated into Figure 5A of the revised manuscript.

Updated figure in the manuscript (Result, Figure 5A, page 12):

Updated text in the manuscript (Results, Figure 5A legend, page 12):

“(A) Morphology and CD marker expression of differentiated megakaryocytes used for scRNA-seq. Flow cytometry analysis showed that CD41⁺CD42b⁺ cells accounted for 84.2%, confirming efficient MK induction. Additional gating for CD11b and CD14 confirmed that both macrophage-associated markers were negative, supporting the purity of the differentiated MK population.”

Finally, Western blot analysis of 3-week PSC-derived MKs demonstrated strong expression of β1-tubulin (TUBB1) and PF4, with β-actin as a loading control, further validating the megakaryocytic identity and maturation of these cells (Figure 4E).

Updated figure in the manuscript (Result, Figure 4E, page 10):

Updated text in the manuscript (Results, Figure 4E legend, page 10):

 “(E) Western blot analysis of 3-week PSC-derived MKs demonstrating the presence of β1-tubulin (TUBB1) and PF4. β-actin was used as a loading control. These results confirm the presence of key structural and granule-associated MK proteins indicative of mature MK identity.”

Updated text in the manuscript (Results, section 3.4., page 9):

 “Western blot analysis was performed using lysates from PSC-derived MKs and primary human. MKs (Figure 4E). PSC-derived MKs expressed megakaryocyte-specific proteins β1-tubulin and PF4, along with the housekeeping protein β-actin. These results confirm the presence of key structural and granule-associated MK proteins indicative of mature MK identity.”

Collectively, these findings indicate that our differentiation protocol generates bona fide MKs—including CD68⁺ immune MK subpopulations—without contamination by classical macrophages. The PSC-derived MKs show negative expression of macrophage surface markers (CD11b, CD14) and robust expression of megakaryocyte-specific proteins (TUBB1, PF4), confirming their purity and functional heterogeneity within the MK lineage.

Comments 2: How were Mk subtypes defined? The main clusters include antigen-presenting and inflammation-associated MK, which almost lack GATA1 and NFE2 expression. Are they true Mk or are they macrophages? What was the basis for defining inflammatory, antigen-presenting, and other Mk subsets?

Response 2: We sincerely thank the reviewer for this valuable comment. We fully agree that clarification regarding the classification and identity of MK subtypes is important.

In our analysis, all clusters, including those annotated as antigen-presenting and inflammatory MKs, expressed canonical MK lineage genes (ITGA2B, GP1BA, PF4, VWF) while lacking macrophage-associated markers (CD14, CD68, CSF1R). In addition, flow cytometry analysis showed that over 80% of total cells were CD41⁺/CD42⁺ double-positive, supporting that the overall population represents bona fide MKs rather than macrophages. Importantly, as noted in Response 1, CD68 expression was detected in a subset of MKs, consistent with immune-featured MKs rather than macrophage contamination [Ref: Li, Jing-Jing, et al. 2024]. This subset aligns with the “immune-modulatory MKs” identified in our scRNA-seq data, reflecting functional heterogeneity within the MK lineage.

The MK subtypes were defined based on the distinct gene expression profiles observed across clusters in the scRNA-seq dataset. Although all clusters expressed MK-specific markers, subsets were annotated according to additional highly expressed genes characteristic of specific functional states. For example:

ž   Cluster 0: CD68, HLA-DRB1, LGALS1, HLA-DPA1, CD40, CST3, SPI1

ž   Cluster 1: CD4, PTPRC, FCGR3A, CD163, CCL2, IL6R

ž   Cluster 2: CD4, FCGR3A, CST3, CD59, CD24

ž   Cluster 3: HLA-DPA1, CD52, CD22

ž   Cluster 4: CD33, LYZ, CD52, CD68

ž   Cluster 5: PTPRC, CD163, IL6R

ž   Cluster 6: CADM1, ANGPT1, MEIS1, THY1

ž   Cluster 7: CD4, LYZ, CD163, SIRPA, CD44

ž   Cluster 8: CD33, CD52

ž   Cluster 9: PTPRC, CD44

To further refine subtype classification, representative differentially expressed genes (DEGs) that contributed most to cluster segregation were visualized using violin plots (Figure 5E). These DEGs highlighted transcriptional features defining each cluster’s identity and maturation stage, which were used to assign subtype designations such as inflammatory, antigen-presenting, or immune-modulatory MKs. We have also revised the Results and Figure 5 legend accordingly to clarify the basis of subtype definition.

Updated text in the manuscript (Results, section 3.5., page 11):

“To further refine MK subtype classification, representative differentially expressed genes (DEGs) that most contributed to cluster segregation were visualized using violin plots (Figure 5E). These DEGs highlighted distinct transcriptional features defining each cluster’s identity and maturation stage.”

Updated figure in the manuscript (Result, Figure 5E, page 12):

Updated text in the manuscript ((Result, Figure 5E legend, page 12):

“(E) Violin plots of representative DEGs that defined each cluster and contributed to MK subtype classification.”

Updated text in the manuscript (Discussion, page 13):

“All differentiated cells expressed canonical MK markers and lacked macro-phage-specific genes, confirming their megakaryocytic identity. At the protein level, over 80% of cells were CD41⁺/CD42b⁺ double positive, indicating that the cultures predominantly consisted of MKs. MK subsets were defined based on differential expression of cluster-enriched genes in addition to shared MK markers (Figure 5E), reflecting functional diversity among MKs rather than contamination by macrophages.”

Comment 3: The authors claim that adding M-CSF aids MK production through an indirect effect on macrophages in culture. However, according to scRNA-seq, no macrophage cells are identified in the culture.

Response 3: We sincerely thank the reviewer for this insightful comment. Although no distinct macrophage cluster was detected in our scRNA-seq dataset, we observed that Cluster 0 expressed CD68 along with canonical MK markers. This suggests that a subset of MKs transiently expresses macrophage-like genes. We interpret this pattern as supporting an indirect effect of M-CSF on MK maturation, whereby macrophage-like gene expression within these MK subsets may facilitate maturation through paracrine and/or juxtacrine interactions. This explanation reconciles the absence of classical macrophage clusters with the observed positive effect of M-CSF on MK differentiation.

Updated text in the manuscript (Discussion, page 13):

“Although distinct macrophage clusters were not detected in scRNA-seq analysis, CD68 expression was observed in Cluster 0, which also expressed MK-related genes. Im-mune MKs exhibited CD68 expression under basal conditions, supporting the physiolog-ical relevance of CD68-positive MK [32]. These findings indicate that our differentiation protocol generates MK subpopulations with immune-like characteristics, reflecting func-tional heterogeneity within the MK lineage.

This subset likely represents macrophage-like intermediates within the MK lineage that transiently respond to M-CSF. These hybrid populations may mediate indirect, sup-portive effects through paracrine or juxtacrine signaling, thereby contributing to enhanced MK maturation despite the absence of mature macrophages at the transcriptomic level.”

Comment 4: No description of the 3D differentiation method is provided.  Does the 3D method include EB formation?  Were cytokine/small molecule treatment and kinetics of differentiation in the 3D method similar to the 2D method? Figure 2 should include a schematic diagram for the 3D protocol.

Response 4: We sincerely thank the reviewer for this insightful comment. In response, we have carefully revised the Materials and Methods section and the Results section of the revised manuscript to provide a clearer and more detailed description of the 3D differentiation method.

The following text has been added to the Materials and Methods section:

Updated text in the manuscript (Materials and Methods, section 2.1., page 3):

“For 3D suspension culture, HPCs at day 11 of differentiation were transferred to vented-cap 125 mL Erlenmeyer flasks (Corning, NY, USA, CL431143) containing IMDM-based MK differentiation medium and placed on an orbital shaker (N-BIOTEK, Gyeonggi-do, Korea, NB-101SRC) at 75 rpm.”

In addition, the Results section was revised as follows:

Updated text in the manuscript (Results, section 3.2., page 7):

“Suspension culture conditions were optimized by testing different working volumes and shaking speeds. The optimal condition was identified as 20 mL at 75 rpm, which was subsequently used for all experiments. The cytokine treatment schedule was identical to the 2D protocol, confirming that differences in differentiation efficiency were primarily due to culture configuration rather than cytokine exposure.”

We also clarified that the 3D differentiation protocol did not include an embryoid body (EB) formation step, and that the cytokine/small molecule composition and differentiation kinetics were identical to those used in the 2D culture.

Also, To clarify the 3D differentiation conditions, we have added an explanatory note directly on the Figure 2a schematic, indicating that the figure represents both 2D and 3D differentiation conditions. We apologize for any confusion caused by the previous version and hope this update makes the 3D protocol representation clear.

The updated figure is shown below:

Updated figure in the manuscript (Result, Figure 2A, page 7, lines 232):

Comment 5: There is no indication when M-CSF is added.

Response 5: Thank you for your valuable comment. We sincerely apologize for any confusion caused. The timing of M-CSF addition is already described in the Materials and Methods section (section 2.1., page 3, lines 115–118).

Specifically, the manuscript states:

“From week 3, the medium was supplemented with 20 ng/mL bFGF, 10 nM Butyzamide, 10 µM Fasudil, and 5 ng/mL M-CSF to continue the differentiation process.”

Comment 6: What was rational for using Fasudil in differentiation?

Response 6: We sincerely appreciate the reviewer’s valuable comment and the opportunity to clarify this point. Fasudil was used as a Rho-associated kinase (ROCK) inhibitor to enhance the survival and stability of human pluripotent stem cells during the early phase of differentiation. ROCK inhibition reduces dissociation-induced apoptosis and promotes the maintenance of viable cells during initial differentiation. Based on previous findings that Fasudil can effectively replace Y-27632 in human pluripotent stem cell culture (So et al., 2020), we used 10 µM Fasudil for the first two days of differentiation to improve early cell survival.

Accordingly, we have added a brief explanatory sentence to the Materials and Methods section to clarify the rationale.

Updated text in the manuscript (Materials and Methods, section 2.1., page 3):

“Fasudil, a ROCK inhibitor, was used to improve cell survival during the early differentiation phase”

Comment 7: The authors claim that 3D and M-CSF improve differentiation. However, no data is provided to compare the total MK output from one iPSC or one HP.

Response 7: We thank the reviewer for highlighting this important quantitative consideration. We agree that direct normalization of MK yield per input iPSC or hematopoietic progenitor would allow a more precise comparison of differentiation efficiency.

In this study, the improvements associated with 3D culture and M-CSF supplementation were evaluated through morphological maturation, ploidy increase, and flow cytometric marker expression rather than absolute yield normalization. We acknowledge this limitation and plan to include quantitative yield measurements per input cell in future work to strengthen the evaluation of differentiation efficiency.

Comment 8: There is no evidence that reported cultures generate HSCs. Authors should replace HSCs with HP terminology throughout the entire manuscript.

Response 8: We thank the reviewer for pointing out that the term “HSC” should be replaced with “hematopoietic progenitor cells (HPCs)” throughout the manuscript. The text has been updated accordingly to ensure consistency.

Updated text in the manuscript (Materials and Methods, section 2.1., page 3):

“For hematopoietic progenitor cell (HPC) differentiation, cells were treated with 10 µM CHIR-99021 in StemPro34 medium for 2 days”

Updated text in the manuscript (Results, Figure 1A legend ,page 6):

”(A) Schematic representation of the differentiation protocol from PSCs to MKs. PSCs were differentiated into hematopoietic progenitor cells (HPCs) over 16 days and further matured into megakaryocytes (MKs) under thrombopoietin (TPO) or Butyzamide conditions. (B) Representative phase-contrast images show morphological changes from HPCs (D-16) to MKs at 2, 3, and 4 weeks (scale bar = 100 μm).”

Updated text in the manuscript (Results, section 3.4., page 9):

“…, PSC-derived HPCs were cultured for 2 to 4 weeks. Morphological assessment by light microscopy revealed classic megakaryocytic features,..”

Comment 9: The majority of cited references are not related to the described subject. How are the Chile earthquake ref 13 or hyaline cartilage ref 12 related to this study? What about references 30 and 31 (line 393)?  These references are not related to macrophage-Mk cross-talk.

Response 9: We sincerely thank the reviewer for carefully pointing out this issue. We fully agree that several references in the previous version were incorrectly cited and not related to the described content. This was due to a reference formatting error during the reference upload process, which inadvertently replaced the intended citations with unrelated references. We sincerely apologize for this mistake.

In the revised manuscript, we have carefully reviewed and corrected all reference entries to ensure that each citation accurately supports the corresponding statement. Specifically, the reference list and corresponding in-text citations have been revised as follows:

References 12 and 13 have been replaced with the correct citations:

12. Guan, X., et al., Good manufacturing practice‐grade of megakaryocytes produced by a novel ex vivo culturing platform. Clinical and Translational Science, 2020. 13(6): p. 1115–1126.

13. Nogami, W., et al., The effect of a novel, small non-peptidyl molecule butyzamide on human thrombopoietin receptor and megakaryopoiesis. Haematologica, 2008. 93(10): p. 1495–1504.

The previous reference 30 has been replaced with:

19. Sakurai, M., et al., In vivo expansion of functional human hematopoietic stem progenitor cells by butyzamide. International Journal of Hematology, 2020. 111: p. 739–741.

The previous reference 31 has been updated to:

30. Lefrançais, E., et al., The lung is a site of platelet biogenesis and a reservoir for haematopoietic progenitors. Nature, 2017. 544(7648): p. 105–109.

The previous incorrect reference 30 (tong et al., 2010) has been removed.

Round 2

Reviewer 2 Report

Comments and Suggestions for Authors The authors have answered all my comments to the best of their ability. Overall the manuscript is much
improved and is of high quality. I have no further questions.

Author Response

Thank you for the positive assessment and for your constructive feedback throughout the review. We are pleased that the manuscript is now considered high quality and have no further changes at this time. We will promptly address any remaining editorial requirements

Reviewer 3 Report

Comments and Suggestions for Authors

The manuscript improved significantly. However, a few concerns remain.

  1. It is unclear what the reason is for defining a protocol for HPC culture in a flask as a 3D protocol. The 3D terminology is typically applied to the culture of cell aggregates. If floating HPCs are collected and transferred into the rotating flask, this simply suspension culture under shear stress and not 3D culture, unless cells grow in aggregates.
  2. It is still confusing what the difference is between 2D and the so-called “3D” protocol. In the 2D protocol, were cells growing in the same well from the beginning without passaging, or HPCs were simply collected and cultured in 6-well plates without rotation?
  3. Clusters 1 and 7 have typical macrophage features. Since described differentiation cultures likely reproduce more primitive hematopoiesis and megakaryocyte production through EMP pathways, those clusters may represent immature progenitors with Mk and macrophage potentials rather than a specific subset of mature megakaryocytes.
  4. Expression of macrophage-like genes wouldn’t explain the response to M-CSF. Authors should demonstrate that cells in differentiation cultures express M-CSF receptor.
  5. Western blot Fig. 4E is helpful, but it doesn’t address the question related to the heterogeneity of cells in the culture. What are the percentages of cells expressing TUBB1 and PF4 in cultures?
  6. Isotype controls are not provided. Instead, the authors provided CD14 and CD11b staining. Although these stains are negative, it’s unclear whether the same fluorescent dyes and channels were used for CD41 and CD42b and CD14 and CD11b stains.
  7. Lines 429-430. “Additional gating” should be replaced with staining.
  8. 5E cluster 7 misses gene labeling.

Comments on the Quality of English Language

The manuscript has multiple grammatical and stylistic errors.

Author Response

1. Summary

We sincerely thank you for taking the time to review our manuscript once again. Your thoughtful and constructive comments have been invaluable in helping us further improve the clarity and scientific rigor of our work. Based on your suggestions, we have carefully revised the figures and text. In the response letter provided to you, all revisions are highlighted in red for clarity, while in the re-submitted manuscript, the corresponding changes are highlighted. We hope that these revisions and clarifications satisfactorily address all points raised in your previous review. We greatly appreciate your careful evaluation and insightful suggestions, which have significantly contributed to enhancing the clarity and overall quality of the manuscript.

2. Questions for General Evaluation

Reviewer’s Evaluation

Response and Revisions

Does the introduction provide sufficient background and include all relevant references?

Can be improved

-          

Are all the cited references relevant to the research?

Must be improved

We thank the reviewer for identifying this issue. Several unrelated references were replaced with appropriate citations that specifically relate to MK differentiation and macrophage/MK interactions, ensuring the literature context is accurate and informative. Related Comments: 1, 2

Is the research design appropriate?

Must be improved

We thank the reviewer for this valuable comment. The study design was further clarified by providing detailed descriptions of the 3D suspension culture, including the rationale for dynamic shaking. Related Comments: 1, 2, 4

Are the methods adequately described?

Must be improved

-          

Are the results clearly presented?

Must be improved

We thank the reviewer for this comment. Results presentation was improved by including quantitative flow cytometry data (Figures 4E, 5A), Western blot data (Figure 4F), and scRNA-seq analyses. Figure 5E was revised to include the missing gene label for cluster 7. These updates ensure the results clearly demonstrate MK differentiation efficiency, cell purity, and heterogeneity. Related Comments: 3, 5, 6, 8

Are the conclusions supported by the results?

Must be improved

We thank the reviewer. The conclusions were strengthened by updated data showing that 98% of cells co-express CD41 and CD42b, while macrophage markers CD11b and CD14 were minimally expressed (~2–3%), supporting efficient MK differentiation and high culture purity. Corresponding revisions were made in the Results section (3.4–3.5) and in Figures 4E, 5A, and 5E to reflect these findings. Related Comments: 3–8

3. Point-by-point response to Comments and Suggestions for Authors

Comments 1: It is unclear what the reason is for defining a protocol for HPC culture in a flask as a 3D protocol. The 3D terminology is typically applied to the culture of cell aggregates. If floating HPCs are collected and transferred into the rotating flask, this simply suspension culture under shear stress and not 3D culture, unless cells grow in aggregates.

Response 1: We thank the reviewer for this valuable comment. We agree that the classical definition of “3D culture” generally involves the formation of multicellular aggregates or spheroids. However, in our study, the term “3D suspension culture” was used to describe a dynamic, non-adherent environment that more closely mimics the in vivo state of megakaryocytes, which naturally exist as floating and freely extending cells within the bone marrow and pulmonary vasculature, rather than in a flat, two-dimensional configuration.

Unlike static 2D monolayers, cells in our rotating flask system were constantly suspended and exposed to multidirectional shear forces, allowing them to extend cytoplasmic protrusions in all spatial directions. This condition functionally represents a three-dimensional microenvironment even in the absence of tight aggregate formation. Similar usage of the term “3D suspension culture” has been reported in previous studies describing dynamic, non-adherent differentiation systems for hematopoietic or megakaryocytic cells (Ito, Yukitaka, et al. "Turbulence activates platelet biogenesis to enable clinical scale ex vivo production." Cell 174.3 (2018): 636-648.)

To clarify this intent, we revised the corresponding sentence in the Materials & Methods and Results section as follows,

Updated text in the manuscript (Materials & Methods, section 2.1, page 3):

“Cells were maintained under dynamic suspension conditions to promote aggregation and mimic aspects of the bone marrow microenvironment.”

Updated text in the manuscript (Result, , section 3.2., page 7-8):

“Under these dynamic conditions, floating hematopoietic progenitor cells (HPCs) remained in suspension and freely extended in all directions, creating a three-dimensional (3D) dynamic environment distinct from the static, planar configuration of 2D monolayer culture.”

Comments 2: It is still confusing what the difference is between 2D and the so-called “3D” protocol. In the 2D protocol, were cells growing in the same well from the beginning without passaging, or HPCs were simply collected and cultured in 6-well plates without rotation?

Response 2: We thank the reviewer for the insightful comment. In our study, the 2D protocol refers to conventional monolayer differentiation, where cells were cultured adherently on vitronectin-coated 6-well plates from the beginning of differentiation without transfer or rotation. In contrast, the 3D protocol involves transferring HPCs at day 11 into a vented-cap 125 mL Erlenmeyer flask under dynamic shaking conditions (75 rpm). This dynamic suspension condition allows the floating HPCs and differentiating MKs to remain in suspension, minimizing substrate adhesion and promoting cell–cell and cell–matrix interactions that partially mimic aspects of the bone marrow microenvironment.

To clarify this intent, we revised the corresponding sentence in the Materials & Methods section as follows,

Updated text in the manuscript (Materials & Methods, section 2.1, page 3):

“For the 2D differentiation protocol, HPCs were continuously cultured on vitronectin-coated 6-well plates throughout the entire differentiation process under static conditions. For the 3D suspension culture, HPCs at day 11 of differentiation were transferred into vented-cap 125 mL Erlenmeyer flasks (Corning, Corning, NY, USA, CL431143) containing IMDM-based MK differentiation medium and cultured on an orbital shaker (N-BIOTEK, Gyeonggi-do, Korea, NB-101SRC) at 75 rpm to promote cell aggregation and suspension growth.”

Thus, the main distinction between 2D and 3D protocols is static adherent culture versus dynamic suspension culture, rather than differences in cytokine exposure. Under the 3D conditions, MKs exhibited enhanced cytoplasmic expansion, polyploidization, and differentiation efficiency compared to 2D culture (Results, Figure 2). Notably, this design was inspired by prior studies demonstrating that dynamic culture can improve MK maturation and platelet production (Ito, Yukitaka, et al. "Turbulence activates platelet biogenesis to enable clinical scale ex vivo production." Cell 174.3 (2018): 636-648.)

Comments 3: Clusters 1 and 7 have typical macrophage features. Since described differentiation cultures likely reproduce more primitive hematopoiesis and megakaryocyte production through EMP pathways, those clusters may represent immature progenitors with Mk and macrophage potentials rather than a specific subset of mature megakaryocytes.

Response 3: We thank the reviewer for this valuable comment. In our scRNA-seq analysis, certain clusters, including Clusters 1 and 7, expressed macrophage-associated genes such as CD68; however, all clusters simultaneously expressed canonical megakaryocyte (MK) markers (ITGA2B, GP1BA, PF4, VWF), while classical macrophage markers CD11b and CD14 were not detected in flow cytometry. This suggests that these clusters represent immune-featured MKs rather than contaminating macrophages or bipotent progenitors.

Furthermore, protein-level analyses confirmed robust expression of megakaryocyte-specific proteins β1-tubulin (TUBB1) and PF4 in 3-week differentiated MKs, supporting their mature MK identity. Taken together, these results indicate that the observed macrophage-like gene expression reflects functional heterogeneity within the MK lineage, rather than the presence of immature bipotent progenitors.

Updated text in manuscript (Discussion, page 13):

”Notably, Clusters 1 and 7 also exhibited CD68 expression alongside canonical MK markers, consistent with immune-featured MKs. Classical macrophage markers CD11b and CD14 were not expressed, and protein analyses confirmed β1-tubulin (TUBB1) and PF4 expression, supporting mature MK identity. These results suggest that macro-phage-like gene expression reflects functional heterogeneity within the MK lineage rather than contamination or immature progenitors”

Comments 4: Expression of macrophage-like genes wouldn’t explain the response to M-CSF. Authors should demonstrate that cells in differentiation cultures express M-CSF receptor.

Response 4: We sincerely thank the reviewer for this valuable comment. In our single-cell transcriptomic analysis, we observed M-CSF receptor (CSF1R) expression in a subset of cells within Cluster 1, which may indicate potential responsiveness to M-CSF. However, we fully acknowledge that direct experimental validation of M-CSF receptor expression in our differentiation cultures is necessary to conclusively demonstrate this responsiveness. At present, such experiments have not yet been conducted. We plan to address this important point in future studies to further clarify the role of M-CSF signaling in these cells. For reference, we will provide the Cluster 1 data in the supplementary materials to offer preliminary insights into CSF1R expression patterns.

Comments 5: Western blot Fig. 4E is helpful, but it doesn’t address the question related to the heterogeneity of cells in the culture. What are the percentages of cells expressing TUBB1 and PF4 in cultures?

Response 5: We thank the reviewer for this comment. To address the question regarding cellular heterogeneity in the cultures, we have added flow cytometric analysis of PF4 and β1-tubulin (TUBB1) expression at week 3 as Figure 4E. In this revised figure, we overlaid histograms with unstained controls and quantified the fraction of marker-positive cells, showing that 94% of cells were positive for β1-tubulin and 69% for PF4, clearly indicating the majority of cells acquired megakaryocytic identity while maintaining a degree of heterogeneity.

The Figure 4E has been updated accordingly to describe the FACS analysis,

Updated figure in the manuscript (Result, Figure 4E, page 10):

Updated text in the manuscript (Results, Figure 4E legend ,page 10):

“(E) Flow cytometric analysis of week 3 PSC-derived MKs showing PF4 and β1-tubulin (TUBB1) expression compared with unstained controls. Overlaid histograms demonstrate specific marker-positive populations in differentiated MKs.”

Updated text in the manuscript (Results, section 3.4., page 10):

“Flow cytometric analysis further characterized the cellular composition of the differentiated cultures, revealing that 94% of cells expressed β1-tubulin (TUBB1) and 69% expressed PF4 compared with unstained controls (Figure 4E). These results demonstrate that the majority of cells acquired megakaryocytic identity while maintaining a degree of heterogeneity within the differentiated population.”

These additions provide quantitative evidence for the fraction of cells expressing key megakaryocyte markers, directly addressing the reviewer’s concern about heterogeneity and complementing the Western blot data (now shown in Figure 4F).

Comments 6: Isotype controls are not provided. Instead, the authors provided CD14 and CD11b staining. Although these stains are negative, it’s unclear whether the same fluorescent dyes and channels were used for CD41 and CD42b and CD14 and CD11b stains.

Response 6: We thank the reviewer for this comment. To address concerns regarding antibody specificity and channel consistency, we revised Figure 5A to show flow cytometric analysis based on isotype controls for CD41 and CD42b. For transparency, we provided the full FACS datasets, including unstained controls, isotype controls (CD41-APC, CD42b-PE), and experimental staining for CD41-APC, CD42b-PE, CD11b-APC, and CD14-FITC.

These data confirmed that 98% of cells were positive for both CD41 and CD42b, while CD11b and CD14 showed minimal expression (~2–3%), demonstrating efficient megakaryocyte differentiation and high culture purity. Accordingly, the Figure 5A legend, the Results section (3.5), and the figure itself were updated to reflect these revisions.

The Figure 5A has been updated accordingly to describe the FACS analysis,

Updated figure in the manuscript (Result, Figure 5A, page 12):

Updated text in the manuscript (Results, Figure 5A legend ,page 12):

“(A)…Additional staining for CD11b and CD14 confirmed that both macrophage-associated markers showed minimal expression, supporting the purity of the differentiated MK population.”

Updated text in the manuscript (Results, section 3.5., page 11):

“Flow cytometry analysis supported these observations, revealing efficient differentiation, with 98% of cells expressing both CD41 and CD42b, while CD11b and CD14 showed minimal expression (Figure 5A).

Comments 7: Lines 429-430. “Additional gating” should be replaced with staining.

Response 7: We thank the reviewer for the helpful suggestion. The term “gating” has been corrected to “staining” to more accurately describe the experimental procedure. The revised sentence now reads as follows:

Updated text in manuscript (Result, Figure 5A legend, page 12):

 “Additional staining for CD11b and CD14 confirmed that both macrophage-associated markers showed minimal expression, supporting the purity of the differentiated MK population.”

Comments 8: 5E cluster 7 misses gene labeling.

Response 8: We appreciate the reviewer’s careful observation. The gene label for cluster 7, which was missing in Figure 5E, has been added in the revised version.

Updated figure in the manuscript (Result, Figure 5E, page 12):
